# Polyunsaturated fatty acid production by *Yarrowia lipolytica* employing designed myxobacterial PUFA synthases

Katja Gemperlein[1], Demian Dietrich [2], Michael Kohlstedt[2], Gregor Zipf[3], Hubert S. Bernauer[3], Christoph Wittmann[2], Silke C. Wenzel[4] & Rolf Müller [1,4]

Long-chain polyunsaturated fatty acids (LC-PUFAs), particularly the omega-3 LC-PUFAs eicosapentaenoic acid (EPA), docosapentaenoic acid (DPA), and docosahexaenoic acid (DHA), have been associated with beneficial health effects. Consequently, sustainable sources have to be developed to meet the increasing demand for these PUFAs. Here, we demonstrate the design and construction of artificial PUFA biosynthetic gene clusters (BGCs) encoding polyketide synthase-like PUFA synthases from myxobacteria adapted for the oleaginous yeast *Yarrowia lipolytica*. Genomic integration and heterologous expression of unmodified or hybrid PUFA BGCs yielded different yeast strains with specific LC-PUFA production profiles at promising yield and thus valuable for the biotechnological production of distinct PUFAs. Nutrient screening revealed a strong enhancement of PUFA production, when cells were phosphate limited. This represents, to the best of our knowledge, highest concentration of DHA (16.8 %) in total fatty acids among all published PUFA-producing *Y. lipolytica* strains.

[1] Department of Microbial Natural Products, Helmholtz Institute for Pharmaceutical Research Saarland and Helmholtz Centre for Infection Research, Saarbrücken, Germany. [2] Institute for Systems Biotechnology, Saarland University, Saarbrücken, Germany. [3] ATG:biosynthetics GmbH, Merzhausen, Germany. [4] Department of Pharmacy, Saarland University, Saarbrücken, Germany. Correspondence and requests for materials should be addressed to R.M. (email: Rolf.Mueller@helmholtz-hips.de)

Application of the hemiascomycetous yeast *Yarrowia lipolytica* has awakened a strong industrial interest due to its capacity to grow efficiently on hydrophobic substrates (e.g., alkanes, fatty acids, and oils) as a sole carbon source and to produce high amounts of organic acids, especially citric acid[1]. Developments of genetic and cellular tools have contributed to the establishment of *Y. lipolytica* as an amenable host for heterologous protein production[2–4]. Due to its non-pathogenic nature, several processes based on *Y. lipolytica* were categorized as generally recognized as safe (GRAS)[5]. *Yarrowia lipolytica* is classified among the oleaginous yeasts because of its ability to accumulate large amounts of lipids (up to 50% of its cell dry weight (CDW))[6]. The crucial difference between oleaginous and non-oleaginous yeasts becomes evident during cultivation under nitrogen-limiting conditions[7,8]: the carbon flux in non-oleaginous yeasts is directed into synthesis of various polysaccharides, whereas in oleaginous yeasts, it is preferentially channeled towards lipid biosynthesis, leading to an accumulation of triacylglycerols within discrete intracellular lipid bodies.

Obviously, *Y. lipolytica* is a promising host strain for recombinant production of long-chain polyunsaturated fatty acids (LC-PUFAs), such as eicosapentaenoic acid (EPA, 20:5, *n*-3) and docosahexaenoic acid (DHA, 22:6, *n*-3), which have beneficial effects on health by lowering triglyceride concentrations and blood pressure[9]. As already demonstrated in a previous study, expression of a bifunctional $\Delta^{12}/\omega3$ desaturase from *Fusarium moniliforme* in *Y. lipolytica* yielded α-linolenic acid (18:3, *n*-3), a precursor of EPA and DHA, at a concentration of 28.1% of total fatty acids (TFAs)[10]. Production of 20.2% γ-linolenic acid (GLA, 18:3, *n*-6) of TFAs was achieved by overexpressing $\Delta^6$ and $\Delta^{12}$ desaturases from *Mortierella alpina*[11]. A *Y. lipolytica* strain that

produces EPA at 56.6% of TFAs was generated by DuPont (USA)[12]. These production levels were reached by overexpression of 30 copies of nine different, codon-optimized genes (20 desaturase genes, eight elongase genes, and two cholinephosphotransferase genes) combined with disruption of four genes, including a gene encoding a peroxisomal biogenesis factor and two genes involved in the lipid metabolism, in the yeast genome. Similarly, DuPont (USA) engineered a *Y. lipolytica* strain capable of producing 18.3% *n*-3 docosapentaenoic acid (DPA, 22:5) of TFAs starting from EPA with an introduced $C_{20/22}$ elongase gene and 5.6% DHA of TFAs proceeding from *n*-3 DPA with an introduced $\Delta^4$ desaturase gene[13].

All these examples are based on the biotransformation of endogenously supplied fatty acids using enzymes from the aerobic PUFA biosynthetic pathways, which employ saturated fatty acids synthesized by the native fatty acid synthase as substrates. In contrast, iteratively acting, multifunctional polyketide synthase (PKS)-like PUFA synthases as found in myxobacteria enable de novo LC-PUFA biosynthesis from acyl-CoA precursors in a multistep process[14] (Fig. 1a). These multienzyme complexes are encoded by PUFA (*pfa*) biosynthetic gene clusters (BGCs)[15]. The main PUFA produced by the myxobacterium *Aetherobacter fasciculatus* (SBSr002) represents DHA[16], whereas arachidonic acid (AA, 20:4, *n*-6) is the predominant PUFA in the myxobacterium *Minicystis rosea* (SBNa008)[17] (Fig. 1b). In this study, synthetic biology techniques were applied to establish heterologous expression systems for myxobacterial PUFA synthases in the evolutionary unrelated host *Y. lipolytica*. Artificial *pfa* BGCs encoding a PUFA synthase plus a 4′-phosphopantetheinyl transferase (PPTase) from *A. fasciculatus* (SBSr002)[14] or *M. rosea* (SBNa008) were redesigned and holistically optimized for *Y.*

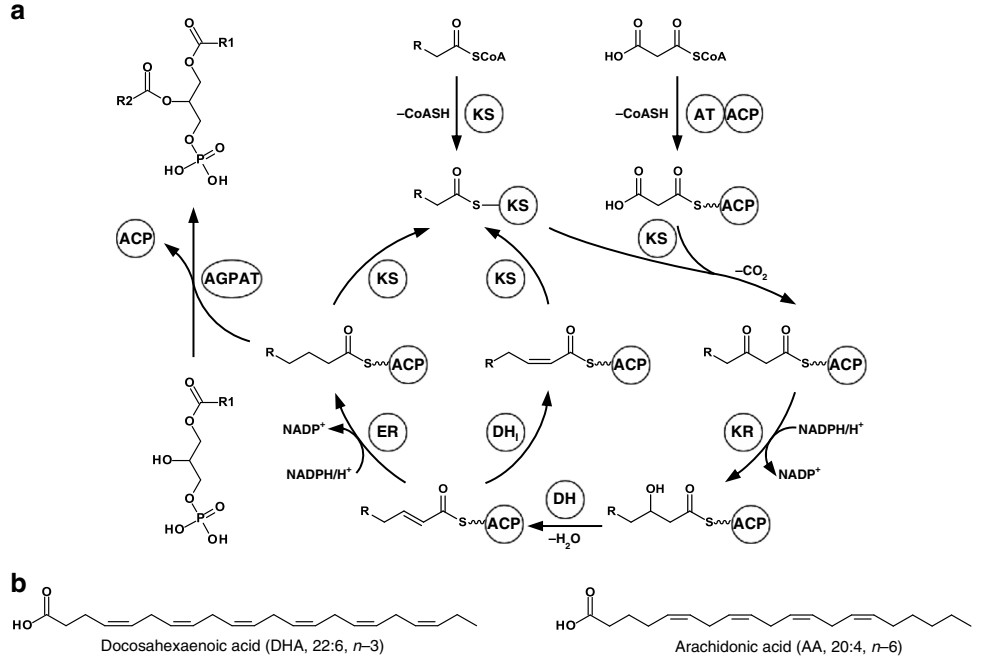

**Fig. 1** De novo polyunsaturated fatty acid (PUFA) biosynthesis in myxobacteria by iteratively acting, multifunctional PUFA synthases. **a** The starter unit acetyl-CoA (R = H) is consecutively elongated with the extender unit malonyl-CoA by several rounds of decarboxylative Claisen condensations, resulting in the extension of the fatty acyl chain by two carbons per cycle. After each round of elongation, the *β*-keto group is either fully reduced by ketoreduction, dehydration plus enoylreduction, or only reduced by ketoreduction and dehydration, giving rise to the *trans* double bond, which is then isomerized to synthesize an acyl chain bearing methylene-interrupted *cis* double bonds. After reaching its final length, the fatty acyl chain is presumably used for acylation of the 2-position of 1-acylglycerol-3-phosphate. KS, ketosynthase of Pfa2 and Pfa3; AT, acyltransferase of Pfa2 and Pfa3; ACP, acyl carrier protein of Pfa2; KR, ketoreductase of Pfa2; DH, dehydratase of Pfa2 and Pfa3; DH$_I$, dehydratase/isomerase of Pfa3; ER, enoylreductase of Pfa1; AGPAT, 1-acylglycerol-3-phosphate O-acyltransferase of Pfa3. **b** Structures of the main PUFAs produced by the myxobacteria *Aetherobacter fasciculatus* (SBSr002) and *Minicystis rosea* (SBNa008)

*lipolytica*, including codon bias adaptation, followed by synthesis and assembly of the respective DNA building blocks. Moreover, diverse hybrid *pfa* BGCs were constructed from these two pathways. Chromosomal integration of the synthetic BGCs yielded transgenic *Y. lipolytica* strains that specifically produce LC-PUFAs, such as AA, EPA, DPA, and DHA.

## Results

**Design of a synthetic BGC encoding a DHA-type PUFA synthase.** The establishment of heterologous expression platforms for recombinant LC-PUFA production using the DPA/DHA-type *pfa* BGC from *A. fasciculatus* (SBSr002) in earlier studies[14,18] served as proof of principle and paved the way for the development of a phylogenetically more distant host organism, exhibiting several advantageous attributes. In this context, the oleaginous yeast *Y. lipolytica* was focused on as expression host of choice, since, in contrast to the myxobacterial LC-PUFA producers, it features GRAS designation, fast growth characteristics, accessibility for genetic manipulation, and ability to accumulate large amounts of lipids. For the heterologous expression of the *pfa* BGC encoding the DPA/DHA-type PUFA synthase plus a PPTase from *A. fasciculatus* (SBSr002)[14,16] in the oleaginous yeast *Y. lipolytica*, synthetic versions of the three *pfa* genes and *ppt* were created (→ BGC version C1_V1; Fig. 2a). Each coding sequence is flanked by the strong hybrid hp4d promoter[3] and the *LIP2* terminator for *Y. lipolytica* to construct single transcription units. Furthermore, non-coding sequences were attached, connecting each transcription unit by 200 bp intergenic linkers. During sequence modulation processes, the myxobacterial genes were subjected to the algorithms of the proprietary *evo*MAG software by ATG:biosynthetics[19]. The software applies concepts of genetic evolutionary algorithms[20,21] to generate sequences that take predefined multivariate sequence parameter values like codon usage frequencies, RNA secondary structures, and sequence motifs into account. Degeneracy of the genetic code allows for the substitution of synonymous codons by silent mutations to modulate the artificial sequence without altering the native amino acid sequence for formal functional biodesigns. The sequence of BGC C1_V1 was designed with the intention to improve the expression by altering the translational elongation profile. An artificial codon usage table was generated from *Y. lipolytica* genome reference data by excluding codons below a predefined synonymous fraction threshold and subsequent normalization of the codon table. Afterwards, each codon position was varied iteratively according to the artificial codon table until the course of the local codon adaptation index was smoothened, and all predetermined scoring requirements were fulfilled (details of the sequence design can be found in Supplementary Note 1, Supplementary Figs. 1, 3, 6, 7, and 9–11, and Supplementary Tables 1, 4–8, 14–19). For practical reasons, the 20.2 kb BGC was dissected into smaller DNA fragments, which were supplied by gene synthesis companies. Constructive sequence requirements for pathway assembly from the synthetic DNA building blocks and for future interchangeability of inter- and intragenic regions were specified and implemented by insertion of unique restriction enzyme sites at any required position within the sequence (Fig. 2a). In parallel, interfering restriction sites had to be eliminated from the sequences.

**Random genomic integration of the artificial *pfa* BGC.** The synthetic pathway C1_V1 encoding PUFA synthases plus a PPTase from *A. fasciculatus* was reconstituted by assembly of the respective DNA building blocks. For the final expression construct, a plasmid backbone was constructed on the basis of the *Y. lipolytica* shuttle vector pINA1312 to enable random integration of typically one copy of the transgenes into the genome of *Y. lipolytica*[2]. The yeast

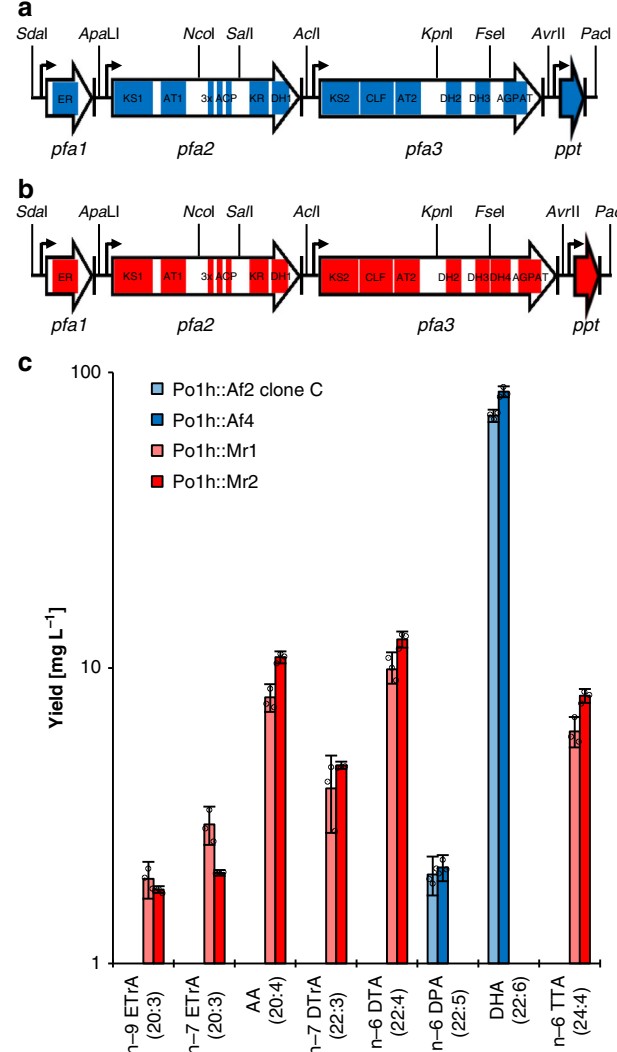

**Fig. 2** Artificial *pfa* biosynthetic gene clusters encoding myxobacterial PUFA synthases for LC-PUFA production in *Yarrowia lipolytica* Po1h. **a** Synthetic *pfa* BGC (20.2 kb) version C1_V1 or C1_V2 comprising genes *pfa1*, *pfa2*, and *pfa3* encoding the DPA/DHA-type PUFA synthase plus gene *ppt* encoding the 4′-phosphopantetheinyl transferase (PPTase) from *Aetherobacter fasciculatus* (SBSr002) adapted for the oleaginous yeast *Yarrowia lipolytica*. **b** Synthetic *pfa* BGC (21.1 kb) version C3 or C3_mod 5′ comprising genes *pfa1*, *pfa2*, and *pfa3* encoding the AA/DTA-type PUFA synthase plus gene *ppt* encoding the PPTase from *Minicystis rosea* (SBNa008) adapted for *Y. lipolytica*. **c** LC-PUFAs produced by *Y. lipolytica* Po1h::Af4 (harboring synthetic *pfa* BGC version C1_V1), *Y. lipolytica* Po1h::Af7 (harboring synthetic *pfa* BGC version C1_V2), *Y. lipolytica* Po1h::Mr1 (harboring synthetic *pfa* BGC version C3), and *Y. lipolytica* Po1h::Mr2 (harboring synthetic *pfa* BGC version C3_mod 5′). Each coding sequence of all clusters is flanked by the strong hybrid hp4d promoter[3] and the *LIP2* terminator. Unique restriction enzyme sites are present at specific positions for pathway assembly and for interchangeability of inter- and intragenic regions. ER, enoylreductase; KS1+KS2, ketosynthases; AT1+AT2, acyltransferases; ACP, acyl carrier protein; KR, ketoreductase; DH1+DH4, polyketide synthase (PKS)-like dehydratases; CLF, chain length factor; DH2+DH3, FabA-like dehydratases/isomerases; AGPAT, 1-acylglycerol-3-phosphate *O*-acyltransferase. Cultivations were carried out in 10 mL YNBG medium+50 mM potassium phosphate buffer pH 6.8 at 28 °C and 200 rpm for 168 h. The indicated values are means and s.d. of three biological replicates, presented on a logarithmic scale. Source data are provided as a Source Data file. ETrA, eicosatrienoic acid; AA, arachidonic acid; DTrA, docosatrienoic acid; DTA, docosatetraenoic acid; DPA, docosapentaenoic acid; DHA, docosahexaenoic acid; TTA, tetracosatetraenoic acid

cassette (DNA sequence devoid of bacterial sequences) of the resulting plasmid pAf2 was liberated via hydrolysis by a restriction endonuclease and transferred into *Y. lipolytica* Po1h[4]. In an initial screening, 38 transformants of *Y. lipolytica* Po1h::Af2 were cultivated in triplicates and their fatty acid methyl esters (FAMEs) were isolated by direct transesterification as well as analyzed by gas chromatography-mass spectrometry (GC-MS). Eleven clones were shown to produce DHA, with transgenic strain *Y. lipolytica* Po1h::Af2 clone C as the best producer. It was re-cultivated in triplicates in 10 mL YNBG medium (heterologous LC-PUFA production was superior in minimal YNBG medium compared to rich, complex YPD medium) at 28 °C for 168 h, and the FAMEs were isolated and analyzed. DHA was produced at a concentration of 71.4 mg L$^{-1}$ or 9.5% of TFAs and 9.8 mg g$^{-1}$ CDW, and also some minor amounts of *n*-6 DPA and *n*-3 DPA were produced (Fig. 2c). Shotgun genome sequencing of *Y. lipolytica* Po1h::Af2 clone C revealed the hypothetical gene YALI0_C05907g as integration site for construct pAf2. These results were a proof of concept for the heterologous expression of artificial *pfa* BGCs in *Y. lipolytica* and served as basis for further engineering of the LC-PUFA expression platform.

**Site-specific genomic integration of the artificial *pfa* BGC.** Gene expression in eukaryotes is a complex process regulated at multiple levels. Transcription of transgenes is not only influenced by recombinant promoters and regulatory DNA elements but also by the spatial positioning of the transgenes within the genome[22,23]. In the course of the present study, locus YALI0_C05907g has emerged as an integration site that enables a good expression of recombinant *pfa* BGCs. Therefore, plasmid pKG2-PIS, allowing for site-specific integration of the *pfa* BGC into locus YALI0_C05907g via double crossover, was constructed. The backbone of the plasmid pAf2 was exchanged for plasmid pKG2-PIS, and the yeast cassette of the resulting plasmid pAf4 was liberated via hydrolysis by restriction endonucleases and transferred into *Y. lipolytica* Po1h[4]. After genotypic verification, transgenic strain *Y. lipolytica* Po1h::Af4 was cultivated in triplicates as described above, followed by isolation and analysis of the FAMEs. As expected, PUFA production was achieved at a similar level (even slightly higher) as observed for *Y. lipolytica* Po1h::Af2 clone C: DHA was detected at a concentration of 86.1 mg L$^{-1}$ or 10.5% of TFAs and 12.8 mg g$^{-1}$ CDW, and *n*-6 DPA plus *n*-3 DPA in some minor amounts were also detected (Fig. 2c and Supplementary Table 9).

Besides the sequence design for BGC C1_V1, we aimed at testing the effect of an alternative gene optimization approach, a codon harmonization resembling strategy, on recombinant LC-PUFA production in *Y. lipolytica*. Therefore, BGC version C1_V2, encoding PUFA synthases plus a PPTase from *A. fasciculatus*, was designed. The constructional sequence design and the assembly of the building blocks were performed analogous to BGC C1_V1 (details of the sequence design can be found in Supplementary Note 1). Interestingly, *Y. lipolytica* Po1h::Af7, containing BGC C1_V2, produces DHA, *n*-6 DPA, and *n*-3 DPA in comparable amounts as *Y. lipolytica* Po1h::Af4 (Supplementary Table 9).

**Design of a synthetic BGC encoding a DTA-type PUFA synthase.** In addition to the artificial *pfa* BGCs encoding the DPA/DHA-type PUFA synthase from *A. fasciculatus*, the *pfa* BGCs encoding PUFA synthases plus a PPTase originating from *M. rosea* (SBNa008)[17] was designed for heterologous expression in the oleaginous yeast *Y. lipolytica*. The constructional and functional sequence design of the resulting 21.1 kb *pfa* BGC C3 (Fig. 2b) was performed analogous to the design of BGC C1_V1 (details of the sequence design can be found in Supplementary

Note 1, Supplementary Figs. 2, 4, and 8, and Supplementary Tables 1 and 2). Six building blocks containing the three *pfa* coding sequences as well as the *ppt* coding sequence flanked by hp4d promoters and *LIP2* terminators, plus intergenic linker sequences were synthesized, assembled, and finally cloned into plasmid pKG2-PIS. The yeast cassette of the resulting plasmid pMr1 was transferred into *Y. lipolytica* Po1h and clones with correct integration of the transgenes into the preferred integration site (YALI0_C05907g) were identified. After three independent cultivations of strain *Y. lipolytica* Po1h::Mr1 as described above, the FAMEs were isolated and analyzed. The transgenic strain produces mainly *n*-6 LC-PUFAs, such as AA (8.0 mg L$^{-1}$), *n*-6 docosatetraenoic acid (DTA, 22:4) (9.9 mg L$^{-1}$), and *n*-6 tetracosatetraenoic acid (TTA, 24:4) (6.1 mg L$^{-1}$) (Fig. 2c and Supplementary Table 9). Calculations of the opening energies within the translation initiation sites of genes *pfa1*, *pfa2*, *pfa3*, and *ppt* of BGC C3 revealed the potential for improvement of the ribosomal access to the translational initiation region on the mRNA level. Consequently, the 5′ coding regions of all four genes were redesigned (all details of the sequence design can be found in the Supplementary Note 1, Supplementary Fig. 5, and Supplementary Tables 2 and 3). DNA fragments carrying the calculated silent mutations in the 5′ coding sequences of the genes were synthesized, and the 5′ coding regions of all the genes of BGC C3 were exchanged for the newly adapted sequences, yielding BGC C3_mod 5′. After transfer of the yeast cassette of the generated plasmid pMr2 into the preferred integration site (YALI0_C05907g) in the genome of *Y. lipolytica* Po1h, strain *Y. lipolytica* Po1h::Mr2 was cultivated in triplicates as described above. Subsequently, the FAMEs were isolated and analyzed. Compared to strain *Y. lipolytica* Po1h::Mr1, LC-PUFA production yields could be indeed increased by 22% using *Y. lipolytica* Po1h::Mr2 (10.9 mg L$^{-1}$ AA, 12.5 mg L$^{-1}$ *n*-6 DTA, and 8.1 mg L$^{-1}$ *n*-6 TTA; Fig. 2c and Supplementary Table 9). The product specificity of the PUFA synthase from *M. rosea* differs widely from that of *A. fasciculatus* with respect to chain length as well as the number and position of double bonds of the produced LC-PUFAs, although the structure of both *pfa* BGCs is very similar (Fig. 2). The most obvious difference between these two multienzyme complexes is the presence of a PKS-type DH domain (DH4) in Pfa3 from *M. rosea*, which is absent in the homologous protein from *A. fasciculatus*.

**Heterologous expression of hybrid *pfa* BGCs.** The factors determining the nature of the produced LC-PUFAs within the (myxo)bacterial PUFA synthases have not yet been elucidated. The production of hybrid PUFA synthases or of PUFA synthases with functional knockouts of domains can contribute to the dissection of the molecular basis of the specificity of PUFA synthase-catalyzed reactions. Consequently, several versions of chimeric *pfa* BGCs were constructed by using the artificial DPA/DHA- and AA/DTA-type *pfa* BGCs. The yeast cassettes of the resulting plasmids described below were transferred into *Y. lipolytica* Po1h and clones with correct integration of the transgenes into the preferred integration site (YALI0_C05907g) were identified. After cultivation of the generated *Y. lipolytica* strains in triplicates as described above, the FAMEs were isolated and analyzed.

At first, the effect of the exchange of the ER domain encoded by *pfa1* on the PUFA production profile was examined. Therefore, plasmid pHyb7, encoding a chimeric PUFA synthase consisting of protein Pfa1 of the DPA/DHA-type PUFA synthase from *A. fasciculatus* and proteins Pfa2 and Pfa3 of the AA/DTA-type PUFA synthase from *M. rosea*, was constructed. The LC-PUFA species produced by this hybrid PUFA synthase are highly

similar to those produced by the AA/DTA-type PUFA synthase of *M. rosea*. Obviously, the nature of the produced LC-PUFAs is not influenced by the ER domain encoded by *pfa1*. Next, the effect of the exchange of the gene *pfa3* or of selected catalytic domains encoded by *pfa3* was evaluated. The exchange of the entire gene *pfa3* from the DPA/DHA-type *pfa* BGC for gene *pfa3* from the AA/DTA-type *pfa* BGC (→ plasmid pSynHybPfaPpt1; Fig. 3a) resulted in the predominant production of the LC-PUFAs EPA and *n*-3 DPA, which are (almost) not produced by the two native PUFA synthases (Fig. 3b). The replacement of domains KS2-CLF-AT2-DH2-DH3 from gene *pfa3* of the DPA/DHA-type *pfa* BGC with the homologous domains from gene *pfa3* of the AA/DTA-type *pfa* BGC plus insertion of domain DH4 from gene *pfa3* of the AA/DTA-type *pfa* BGC led to plasmid pHyb2a (Fig. 3a). Plasmid pHyb5a was constructed by the exchange of domains DH2–DH3 from gene *pfa3* of the DPA/DHA-type *pfa* BGC for the homologous domains from gene *pfa3* of the AA/DTA-type *pfa* BGC plus insertion of domain DH4 from gene *pfa3* of the AA/DTA-type *pfa* BGC (Fig. 3a). As in the case of pHyb1, *Y. lipolytica* Po1h transformants containing the hybrid *pfa* BGCs originating either from plasmid pHyb2a or from plasmid pHyb5a produce EPA and *n*-3 DPA as major products (Fig. 3b). Remarkably, the FabA-like DH domains DH2 and DH3 and the PKS-like DH domain DH4 seem to have a big impact on the PUFA product specificity of Pfa3. Subsequently, the effect of the exchange of the AGPAT domain of *pfa3* and/or the insertion of the DH4 domain was investigated. Both the exchange of the AGPAT domain from gene *pfa3* of the DPA/DHA-type *pfa* BGC for the AGPAT domain from gene *pfa3* of the AA/DTA-type *pfa* BGC and the insertion of domain DH4 from gene *pfa3* of the AA/DTA-type *pfa* BGC led to plasmid pHyb6 (Fig. 3c). The resulting strain *Y. lipolytica* Po1h::Hyb6 exhibits production of its main product *n*-3 DPA at a concentration of 48.0 mg L$^{-1}$ or 5.7% of TFAs and 7.0 mg g$^{-1}$ CDW (Fig. 3d). The insertion of only domain DH4 from gene *pfa3* of the AA/DTA-type *pfa* BGC into the DPA/DHA-type *pfa* BGC (→ plasmid pHyb6b; Fig. 3c) altered the product spectrum towards *n*-3 DPA and DHA as major products (Fig. 3d). However, the inactivation of the DH4 domain in plasmid pHyb6b by converting the active site histidine residue into an alanine residue (→ plasmid pHyb6b-H2270A; Fig. 3c) results in a production profile, which is again identical to that of the DPA/DHA-type PUFA synthase from *A. fasciculatus* (Figs. 2c and 3d). Hence, it can be stated that insertion of domain DH4 has a large impact on the LC-PUFA product specificity of Pfa3.

Apart from the investigations on gene *pfa3*, the effect of the exchange of either the entire gene *pfa2* or, based on the observations made with the DH domains of *pfa3*, of the exchange of only domain DH1 was examined. Hence, gene *pfa2* or domain DH1 from the DPA/DHA-type *pfa* BGC was exchanged for the homologous gene or domain from the AA/DTA-type *pfa* BGC. Interestingly, in both cases the product spectrum is altered towards ω-6 LC-PUFAs with *n*-6 DPA as the main product (Fig. 3e, f). Thus, the nature and the ratio of the produced LC-PUFAs are identical either when the complete gene has been replaced (→ plasmid pHyb8; Fig. 3e) or when only domain DH1 has been exchanged (→ plasmid pHyb9; Fig. 3e). It seems that this PKS-type DH domain is one of the domains affecting the preference of Pfa2 for the production of certain LC-PUFA species. Vice versa, the replacement of domain DH1 from gene *pfa2* of the AA/DTA-type *pfa* BGC by the homologous domain from gene *pfa2* of the DPA/DHA-type *pfa* BGC (→ plasmid pHyb15; Fig. 3g) changes the production profile towards ω-3 LC-PUFAs with *n*-3 DPA as the major product (Fig. 3h).

From all of these data, the conclusion can be drawn that the control of chain length and number/position of double bonds seems to require a very complex interplay of multiple functional domains with the DH domains as important interaction partners. From a biotechnological point of view, the expression of hybrid *pfa* BGCs is especially beneficial, as it exhibits the potential for the production of valuable LC-PUFAs, such as EPA and *n*-3 DPA, which are (almost) not produced by the two native PUFA synthases.

**Optimization of DHA production by medium design**. In order to improve overall productivity of PUFA in *Y. lipolytica*, different medium compositions and culture conditions were compared to investigate their impact on key performance criteria of the DHA producer *Y. lipolytica* Po1h::Af4, such as titer, productivity, and selectivity. The basic setup, using glucose together with YNB and a phosphate buffer yielded about 20 mg L$^{-1}$ DHA, which represented about 6% of TFAs (Fig. 4). Additional analyses revealed that glycerol was particularly efficient to drive DHA biosynthesis (Fig. 4a). The DHA titer achieved was about twice as high as compared to glucose. Surprisingly, a restriction in the availability of phosphate turned out to be important for production efficiency. On glucose as well as on glycerol, the DHA titer was significantly higher, when the phosphate buffer was replaced by MES (2-(*N*-morpholino)ethanesulfonic acid). Furthermore, phosphate limitation enhanced biosynthetic selectivity with regard to the total DHA content in the biomass as well as the fraction of DHA among the TFAs (Fig. 4b). A further reduction of phosphate by lower levels of YNB finally enabled the accumulation of more than 100 mg L$^{-1}$ DHA and a high DHA content of 16.6% of TFAs (Fig. 4d). In contrast, nitrogen limitation was not necessarily required, different to the common picture. As an example, poor DHA accumulation resulted when glucose-grown cells early depleted ammonium, whereas the production was high in the presence of excess ammonium during the whole cultivation time (Supplementary Fig. 12). Generally, high-producing conditions resulted in the accumulation of citrate, which was partially re-consumed in later stages of the cultivation. Acetate, tested as an alternative carbon source, triggered an even higher selectivity of 16.8%, but yielded lower levels of biomass (Supplementary Table 10).

**Fed-batch production process for DHA**. To assess performance under industrially relevant conditions, we benchmarked the DHA producer *Y. lipolytica* Po1h::Af4 in a fed-batch process, which was operated under phosphate-limiting conditions (Fig. 5). Glucose and glycerol were tested in parallel setups as carbon source. On both substrates, the strain grew fast during the initial batch phase and reached a biomass level of about 30 g L$^{-1}$. During this phase, DHA was not accumulated. Glycerol catabolism resulted in a slight accumulation of citrate, which was not observed in the glucose-based process. Both substrates were efficiently consumed during the first 48 h. Likely triggered by the limitation of phosphate, growth then stopped and the cells switched into a producing mode. The DHA level increased to more than 350 mg L$^{-1}$ on glucose after 300 h. On glycerol, the final titer was slightly lower (300 mg L$^{-1}$). The glucose-based process was also superior with regard to the fraction of DHA formed. The DHA content gradually increased during the process and reached a final value of >10%. The imposed feed rate enabled low substrate levels during the entire feed phase, which, however, were still high enough to keep still cells in their producing mode. Citrate accumulated to some extent during the feed phase. Other by-products were not formed.

## Discussion

In this project, we established expression systems for recombinant LC-PUFA production in the oleaginous yeast *Y. lipolytica* based

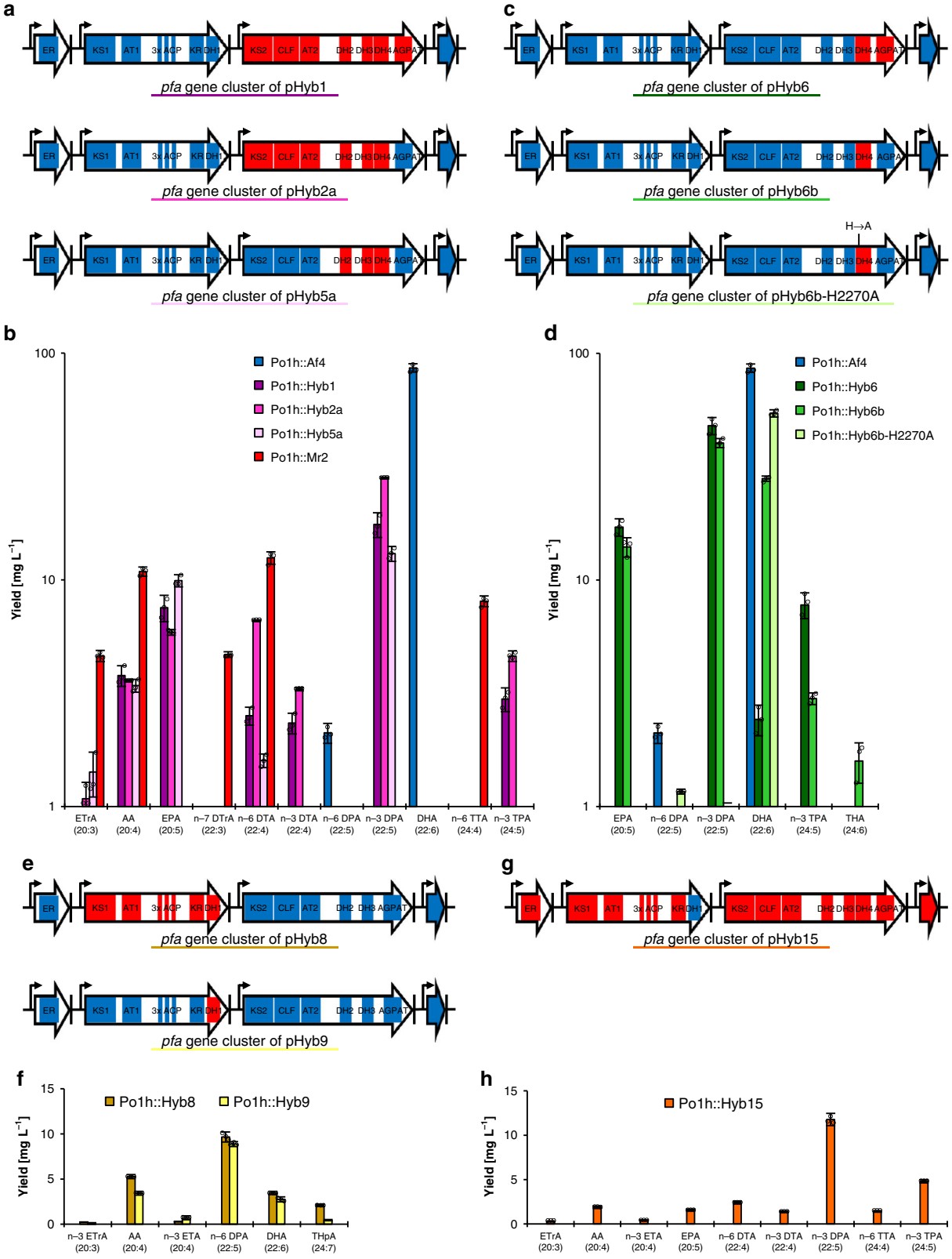

on myxobacterial PUFA biosynthetic machineries, employing methods of Synthetic Biology. By the usage of synthetic, codon-optimized PUFA BGCs encoding homologous PUFA synthases from two myxobacterial species, *Y. lipolytica* production strains with completely different LC-PUFA production profiles regarding chain length as well as the number and position of the double bonds could be generated. Remarkably, one of these *Y. lipolytica*

strains is able to produce the valuable long-chain ω-3 fatty acid DHA at a concentration of 350 mg L$^{-1}$ or 16.8% of TFAs under improved fermentation conditions. DHA is critical for normal development and functioning of the brain[24] and is predominantly present in marine fish and the corresponding fish oils[25], a non-sustainable source. The exchange of single genes or domains—especially the DH domains—between the two synthetic *pfa* BGCs

**Fig. 3** LC-PUFAs produced by *Yarrowia lipolytica* Po1h using chimeric PUFA synthases encoded by artificial *pfa* biosynthetic gene clusters. **a** Hybrid *pfa* BGCs located on plasmid pHyb1, plasmid pHyb2a, and plasmid pHyb5a. **b** LC-PUFAs produced by *Y. lipolytica* Po1h::Hyb1, *Y. lipolytica* Po1h::Hyb2a, and *Y. lipolytica* Po1h::Hyb5a. **c** Hybrid *pfa* BGCs located on plasmid pHyb6, plasmid pHyb6b, and plasmid pHyb6b-H2270A. **d** LC-PUFAs produced by *Y. lipolytica* Po1h::Hyb6, *Y. lipolytica* Po1h::Hyb6b, and *Y. lipolytica* Po1h::Hyb6b-H2270. **e** Hybrid *pfa* BGCs located on plasmid pHyb8 and plasmid pHyb9. **f** LC-PUFAs produced by *Y. lipolytica* Po1h::Hyb8 and *Y. lipolytica* Po1h::Hyb9. **g** Hybrid *pfa* BGCs located on plasmid pHyb15. **h** LC-PUFAs produced by *Y. lipolytica* Po1h::Hyb15. Domains from the synthetic DPA/DHA-type *pfa* BGC are shown in blue; domains from the synthetic AA/DTA-type *pfa* BGC are shown in red. Cultivations were carried out in 10 mL YNBG medium+50 mM potassium phosphate buffer pH 6.8 at 28 °C and 200 rpm for 168 h. The indicated values are means and s.d. of three biological replicates. Source data are provided as a Source Data file. ETrA, eicosatrienoic acid; AA, arachidonic acid; ETA, eicosatetraenoic acid; EPA, eicosapentaenoic acid; DTrA, docosatrienoic acid; DTA, docosatetraenoic acid; DPA, docosapentaenoic acid; DHA, docosahexaenoic acid; TTA, tetracosatetraenoic acid; TPA, tetracosapentaenoic acid; THA, tetracosahexaenoic acid; THpA, putative tetracosaheptaenoic acid

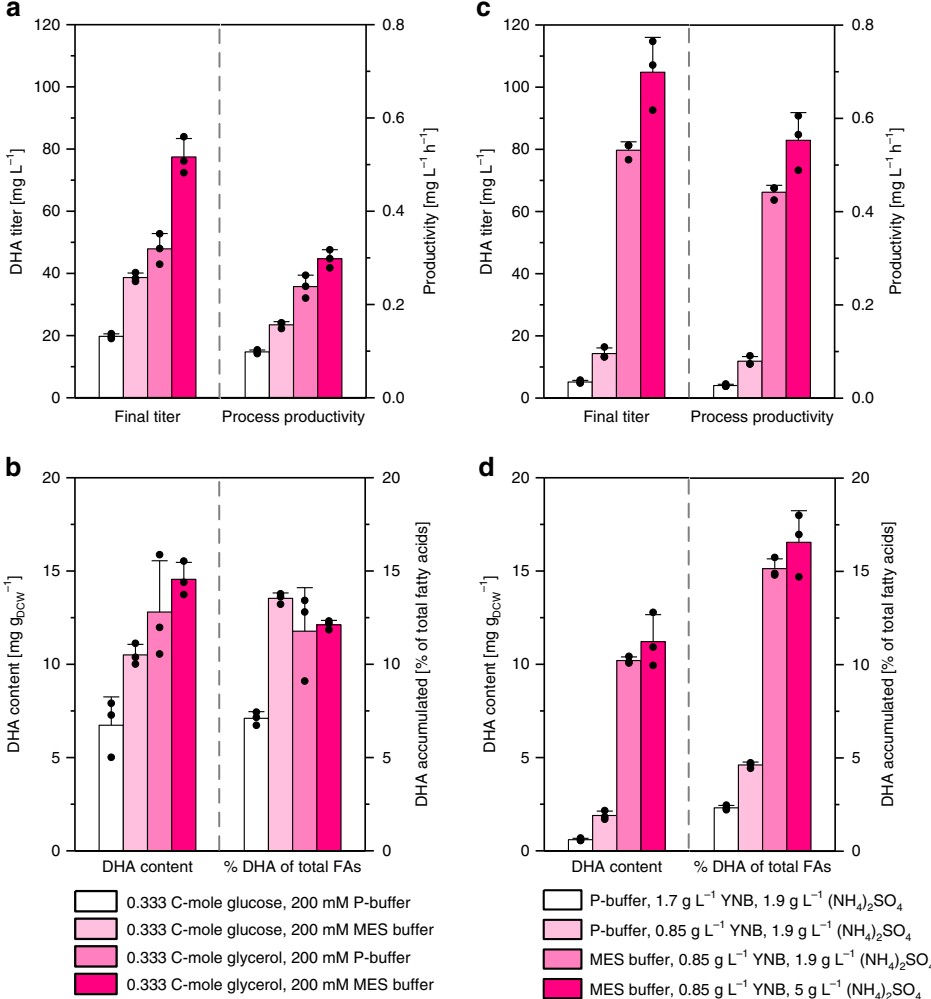

**Fig. 4** DHA production improvement in *Y. lipolytica* Po1h::Af4 based on an optimization of medium composition. Shake flask cultivations were carried out in triplicates. Performance parameters were determined after 200 h of cultivation. Graphs show means and s.d. Source data are provided as a Source Data file

led to further extension of the LC-PUFA product spectrum upon heterologous expression in *Y. lipolytica*. In particular, a *Y. lipolytica* strain producing decent amounts of *n*-3 DPA (48 mg L$^{-1}$/5.7% of TFAs under non-optimized fermentation conditions) was created. This long-chain ω-3 fatty acid was shown to be very beneficial for human health[26], but its concentration in marine fish is substantially lower than those of EPA and DHA[25].

With regard to further development of the LC-PUFA production system established in this study, it will be mandatory to significantly increase the production titers of the LC-PUFAs of interest to reach a cost-effective commercial production. In order to do so, the fermentation procedure has to be further optimized in the matter of media, temperature, duration, and so on, and the amount and the productivity of the (hybrid) PUFA synthases have to be enhanced in vivo. Besides, the genomes of the transgenic *Y. lipolytica* strains can be modified with respect to inactivation or overexpression of genes, which are relevant for production (systems metabolic engineering). As a result, LC-PUFAs should be enriched in the triacylglycerol fraction of the cell and thus increasingly accumulated in intracellular lipid droplets. On the other hand, an adequate supply of the PUFA synthase with the substrates malonyl-CoA and NADPH can be ensured and potential PUFA-degrading reactions can be reduced.

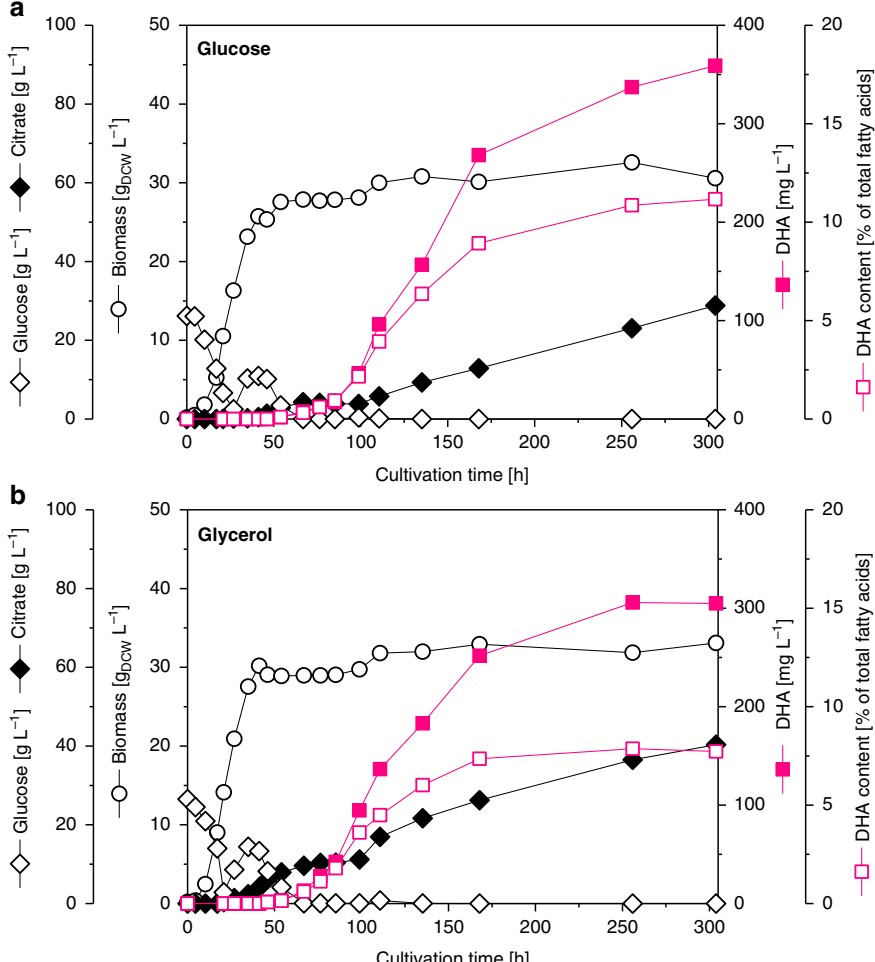

**Fig. 5** DHA production in a fed-batch process. The designed producer *Y. lipolytica* Po1h::Af4 was grown in minimal medium on either glucose or glycerol as sole carbon source. Fermentation was conducted at 28 °C, pH 5.5, and a dissolved oxygen level of 5% during the feed phase. Substrate feeding was initiated upon carbon depletion. Coefficients of variation (CVs) across biological replicates were below 5% for biomass, substrate, and citrate levels, and below 10% for PUFA and native fatty acid content. Source data are provided as a Source Data file

As *Y. lipolytica* has turned out to be an especially suitable microbe for LC-PUFA production, not least because of the ability to accumulate large amounts of lipids[6], industrial efforts from DuPont (USA) on the engineering of *Y. lipolytica* strains capable of producing large amounts of LC-PUFAs were also reported[12,13]. Contrary to the *Y. lipolytica* strains generated in this study, which employ multifunctional PUFA synthases, DuPont made use of alternatingly acting position-specific desaturases and elongases, belonging to the aerobic PUFA biosynthetic pathway. In order to ensure efficient expression of the heterologous genes in *Y. lipolytica*, integration vectors with codon-optimized coding sequences driven by strong *Y. lipolytica* promoters were constructed. In total, three copies of $\Delta^{12}$ desaturase genes, two copies of $\Delta^6$ desaturase genes, four copies of $C_{18/20}$ elongase genes, five copies of $\Delta^5$ desaturase genes, three copies of $\Delta^{17}$ desaturase genes, three copies of $C_{16/18}$ elongase genes, one copy of a $C_{20/22}$ elongase gene, and one copy of a $\Delta^4$ desaturase gene have been integrated into the genome of DuPont's *Y. lipolytica* strain Y3000[13]. Moreover, the acyl-CoA oxidase 3 gene *POX3* and an endogenous $\Delta^{12}$ desaturase had to be inactivated. In addition to the fatty acids produced by the corresponding wild type, strain Y3000 synthesizes 5.6% DHA, 18.3% *n*-3 DPA, 9.7% $C_{20}$ PUFAs, and 30.1% GLA of TFAs. As opposed to this, simply four biosynthetic genes encoding the DPA/DHA-type myxobacterial PUFA synthase and the PPTase from *A. fasciculatus* (SBSr002)

were integrated into the genome of *Y. lipolytica* Po1h. The resulting strain *Y. lipolytica* Po1h::Af4 produces already 10.5% DHA with high selectivity (production of only 0.4% non-preferred PUFAs) under non-optimized fermentation conditions.

Substantial advantages of LC-PUFA biosynthesis via PUFA synthases as opposed to exploitation of the aerobic pathways are the independence from endogenous fatty acids as biosynthetic precursors and the lower consumption of NAD(P)H. For instance, de novo synthesis of DHA catalyzed by PUFA synthases merely relies on 14 NADPH molecules. However, using the aerobic route in which palmitic acid (synthesized by fatty acid synthase using 14 NADPH molecules) is converted into DHA via diverse fatty acid desaturases and elongases, additionally 12 NAD(P)H molecules are consumed. Therefore, the production of LC-PUFAs using PUFA synthases described here should in the future allow for an optimized and highly efficient process towards these valuable compounds.

## Methods

**Sequence analysis and design of synthetic gene clusters**. The sequences of the *pfa* BGCs plus the genes encoding the PPTase from *A. fasciculatus* (SBSr002) and *M. rosea* (SBNa008) were analyzed and compared to the genome sequence of *Y. lipolytica* CLIB 122[27] retrieved from NCBI Genome RefSeq NC_006067, NC_006068, NC_006069, NC_006070, NC_006071, and NC_006072. Based on this, relevant parameters for constructional and functional sequence design were defined to generate artificial pathway versions using the *evo*MAG[is] software

package (ATG:biosynthetics GmbH)[19]. The sequence design process included engineering of restriction sites, adaptation of the codon bias, and elimination of sequence repeats, as well as rare codon clusters, elimination of potential donor splice signals, and introduction of hidden stop codons in unused frames. Further details on the performed sequence analyses and the design of synthetic BGCs can be found in Supplementary Note 1.

**Culture conditions.** *Escherichia coli* DH10B[28] and *E. coli* NEB 10-β (New England Biolabs) were used for cloning experiments. *Escherichia coli* HS996/pSC101-BAD-gbaA (tet[R])[29] were used for modification of a plasmid using Red/ET recombination. The cells were grown in 2xYT medium (1.6% tryptone, 1% yeast extract, 0.5% NaCl), in LB medium or on LB agar (1% tryptone, 0.5% yeast extract, 0.5% NaCl (1.5% agar)) at 30–37 °C (and 200 rpm) overnight. Antibiotics were used at the following concentrations: 50 µg mL$^{-1}$ kanamycin, 34 µg mL$^{-1}$ chloramphenicol, and 6 µg mL$^{-1}$ tetracycline.

Auxotrophic *Y. lipolytica* strain Po1h (CLIB 882)[4] was obtained from Centre International de Ressources Microbiennes (CIRM)-Levures, Institut National de la Recherche Agronomique (INRA), AgroParisTech (Thiverval-Grignon, France). It was grown in YPD medium or on YPD agar containing 1% yeast extract, 2% peptone, and 2% dextrose. Prototrophic transformants were grown on minimal YNB-N$_{5000}$ agar containing 0.67% yeast nitrogen base (with 75% (NH$_4$)$_2$SO$_4$ and without amino acids), 1% glucose, and 1.5% agar or in minimal YNBG liquid medium containing 0.67% yeast nitrogen base (with 75% (NH$_4$)$_2$SO$_4$ and without amino acids), 2% (w/v) glycerol, and 50 mM potassium phosphate buffer, pH 6.8. The cultures were incubated at 28–30 °C.

**Isolation of genomic DNA from *Y. lipolytica*.** Isolation of genomic DNA from yeast cells for genome sequencing was carried out using the method of Hoffman and Winston[30]. In the first step, 50 mL of a culture of *Y. lipolytica* grown at 28 °C were harvested by centrifugation at 12,000 × *g* for 5 min. The supernatant was decanted, and 200 µL lysis buffer (2% (v/v) Triton X-100, 1% sodium dodecyl sulfate, 100 mM NaCl, 1 mM EDTA, and 10 mM Tris-Cl, pH 8.0), 200 µL phenol:chloroform:isoamyl alcohol (25:24:1), plus 0.3 g acid-washed glass beads (425–600 µm) were added to the cells. The tube was vortexed for 3 min. Afterwards, 200 µL TE buffer (10 mM Tris-Cl, pH 8.0, and 1 mM EDTA) were added, and the sample was centrifuged at 21,000 × *g* for 5 min. The upper aqueous phase was transferred to a tube containing 1 mL ice-cold ethanol and mixed by inversion. The sample was centrifuged at 21,000 × *g* for 2 min, and the supernatant was discarded. The pellet was resuspended in 400 µL TE buffer plus 30 µg RNase A. After incubation at 37 °C for 15 min, 44 µL of 4 M ammonium acetate and 1 mL ice-cold ethanol were added, and the tube was inverted to mix. Genomic DNA was precipitated at −80 °C for 30 min to increase yield. The sample was centrifuged at 21,000 × *g* for 2 min, and the supernatant was discarded. The pellet was washed with 700 µL of 70% ethanol, centrifuged at 21,000 × *g* for 1 min, and the supernatant was discarded. The dried DNA was resuspended in 50 µL of 5 mM Tris-Cl, pH 8.0.

**Transformation of *Y. lipolytica*.** Transformation of *Y. lipolytica* was carried out using a protocol developed by M.-T. Le Dall, modified by C. Madzak (Laboratoire de Microbiologie de l'Alimentation au Service de la Santé (MICALIS), AgroParisTech, Thiverval-Grignon, France; personal communication). One loopful of *Y. lipolytica* cells from a YPD agar plate grown at 30 °C overnight were resuspended in 1 mL TE buffer in a sterile tube. The cells were centrifuged at 10,000 × *g* for 1 min, and the supernatant was discarded. After resuspension in 600 µL of 0.1 M lithium acetate, pH 6.0, the cells were incubated at 28 °C for 1 h in a water bath. The samples were centrifuged at 850 × *g* for 2 min, the supernatant was discarded, and the cells were softly resuspended in 80 µL of 0.1 M lithium acetate, pH 6.0. Forty microliters of competent cells were mixed with 2.5 µL herring testes carrier DNA (10 mg mL$^{-1}$ in TE buffer, denatured) and 2.5 µL linear DNA to be transferred. The samples were incubated at 28 °C for 15 min in a water bath, and 350 µL of 40% PEG 4000 in 0.1 M lithium acetate, pH 6.0 (plus 16 µL of 1 M dithiothreitol in case of non-targeted genome integration) were added. After incubation of the cells at 28 °C for 1 h in a water bath, 40 µL dimethyl sulfoxide (DMSO) were added. Subsequently, heat shock was carried out at 39 °C for 10 min in a heating block. Six hundred microliters of 0.1 M lithium acetate, pH 6.0, were then added, and the cells were plated onto YNB-N$_{5000}$ agar. The plates were incubated at 30 °C for 3 days. Thereafter, selected colonies were transferred in 1 mL YNBD or YNBG medium and cultivated at 28–30 °C and 900 rpm. The constructed strains are listed in Supplementary Table 13.

**Construction of an initial plasmid for random integration.** For modification of plasmid pINA1312[2] in order to construct plasmid pINA1312-*Sda*I-cm$^R$-*Pac*I (Supplementary Table 12), a 1.2 kb DNA fragment containing a chloramphenicol resistance gene flanked with *Sda*I and *Pac*I restriction sites as well as the appropriate homology arms was amplified from plasmid pACYC184 (New England Biolabs) using primers HA+*Sda*I+cm$^R$ for pINA1312_fwd and HA+*Pac*I+cm$^R$ for pINA1292+1312_rev (Supplementary Table 11). PCR was performed with *Taq* DNA polymerase (Thermo Scientific) under standard conditions according to the manufacturer's protocol. The reactions contained 8% glycerol and were carried out in an Eppendorf Mastercycler under the following conditions: initial denaturation

for 3 min at 95 °C; 30 cycles consisting of denaturation for 30 s at 95 °C, annealing for 30 s at 62 °C, and extension for 1 min/kb at 72 °C; and a final extension for 10 min at 72 °C. The linear plus circular homologous Red/ET recombineering[14] was performed using 1.5 µL of the PCR product.

**Construction of plasmids for targeted integration.** A cloning plasmid with chloramphenicol resistance gene, p15A origin of replication, and a multiple cloning site comprising *Sgs*I, *Not*I, *Sgr*DI, *Sda*I, *Pac*I, and *Swa*I restriction sites was constructed. This plasmid (2.1 kb) was amplified from plasmid pACYC184 using overlapping primers cm$^R$_*Sda*I+*Pac*I+*Swa*I+*Sgs*I_fwd and p15A_*Sgs*I+*Not*I +*Sgr*DI_rev. PCR was performed with the PCR extender system (5 Prime) under standard conditions according to the manufacturer's protocol. The reactions were carried out in an Eppendorf Mastercycler under the following conditions: initial denaturation for 2 min at 94 °C; 20 cycles consisting of denaturation for 20 s at 94 °C, annealing for 20 s at 66 °C, and extension for 1 min kb$^{-1}$ at 72 °C; and a final extension for 10 min at 72 °C. The *hph* gene (1 kb) encoding the hygromycin B phosphotransferase from *E. coli*/*Klebsiella pneumoniae* was amplified from plasmid Tn5_epo_hyg (Y. Zhang, unpublished) using primers hph_A_fwd and hph_Bam-HI_rev. PCR was performed with Phusion DNA polymerase (Thermo Scientific) under standard conditions according to the manufacturer's protocol. The reactions contained 5% DMSO and were carried out in an Eppendorf Mastercycler under the following conditions: initial denaturation for 2 min at 98 °C; 30 cycles consisting of denaturation for 15 s at 98 °C, annealing for 20 s at 70 °C, and extension for 10 s kb$^{-1}$ at 72 °C; and a final extension for 10 min at 72 °C. The PCR amplicon was inserted into plasmid pINA1312 via *Pml*I and *Bam*HI restriction sites, yielding plasmid pINA1312-*hph*. For the fusion of hp4d-*hph* and *LIP2*t, the strong hybrid promoter hp4d[3] and *hph* plus the 5′ part of the *LIP2* terminator as overlapping sequence were amplified as the first fragment (1.6 kb) from plasmid pINA1312-*hph* using primers UAS1B_*Sda*I_fwd and hph+*LIP2*t overlap_rev. The second fragment (0.1 kb) contained the *LIP2* terminator from *Y. lipolytica* plus the 3′ end of *hph* as overlapping sequence and was amplified from plasmid pACYC_BB1-4_C1_V1 using primers *LIP2*t+hph overlap_fwd and *LIP2*t_*Sgr*DI_rev. PCR amplification of the two fragments to be spliced was performed with Phusion DNA polymerase (Thermo Scientific) under standard conditions according to the manufacturer's protocol. The reactions contained 5% DMSO and were carried out in an Eppendorf Mastercycler under the following conditions: initial denaturation for 2 min at 98 °C; 20 cycles consisting of denaturation for 15 s at 98 °C, annealing for 20 s at 64 °C, and extension for 10 s kb$^{-1}$ at 72 °C; and a final extension for 10 min at 72 °C. For the subsequent overlap extension PCR using primers UAS1B_*Sda*I_fwd and *LIP2*t_*Sgr*DI_rev, the two amplified fragments were used as templates. PCR of the 1.7 kb fragment was performed as described for the amplification of the two fragments to be spliced. Plasmid pKG1 was constructed by ligation of hp4d-*hph*-*LIP2*t with the chloramphenicol resistance gene, p15A origin of replication, and the multiple cloning site via *Sda*I and *Sgr*DI restriction sites. One kilobase homology regions upstream and downstream of the preferred integration site were amplified from genomic DNA of *Y. lipolytica* Po1h::Af2 clone C using primers PIS_fragment1_*Not*I_fwd/PIS_-fragment1_*Pci*I+*Sgr*DI_rev and PIS_fragment2_*Pac*I_fwd/PIS_fragment2_*Swa*I_rev. PCR was performed with Phusion DNA polymerase (Thermo Scientific) under standard conditions according to the manufacturer's protocol. The reactions contained 5% DMSO and were carried out in an Eppendorf Mastercycler under the following conditions: initial denaturation for 2 min at 98 °C; 20 cycles consisting of denaturation for 15 s at 98 °C, annealing for 20 s at 61 °C, and extension for 10 s kb$^{-1}$ at 72 °C; and a final extension for 10 min at 72 °C. The PCR amplicons were inserted into plasmid pKG1 via *Not*I/*Sgr*DI and *Pac*I/*Swa*I restriction sites, yielding plasmid pKG1-PIS. In order to generate plasmid pKG2-PIS, hp4d-*hph*-*LIP2*t of plasmid pKG1-PIS was replaced by the orotidine 5′-phosphate decarboxylase expression cassette from *Y. lipolytica* (*URA3*p-*URA3*-*URA3*t) located on plasmid pINA1312-*Sda*I-cm$^R$-*Pac*I via *Pci*I and *Sda*I restriction sites. Sequences of primers used in this study are listed in Supplementary Table 11 and further information on the plasmids constructed in this study are provided in Supplementary Table 12.

**Cloning of expression plasmids containing synthetic *pfa* BGCs.** For the heterologous expression of artificial PUFA biosynthetic pathways in *Y. lipolytica*, four building blocks containing the three *pfa* genes as well as gene *ppt* encoding the PPTase originating from *A. fasciculatus* (SBSr002) (→ synthetic BGCs version C1_V1 and C1_V2) and six building blocks containing the three *pfa* genes as well as gene *ppt* encoding the PPTase originated from *M. rosea* (SBNa008) (→ synthetic BGC version C3) were designed and supplied by gene synthesis companies in the standard cloning vectors pBSK or pGH. Each coding sequence is flanked by the strong hybrid hp4d promoter[3] plus *LIP2* terminator for *Y. lipolytica*. Moreover, non-coding sequences were attached to each gene allowing for the connection of the transcription units by 200 bp intergenic linkers. In the course of constructional design, the unique *Sda*I, *Apa*LI, *Nco*I, *Sal*I, *Acl*I, *Kpn*I, *Fse*I, *Avr*II, and *Pac*I restriction sites were introduced for cloning purposes as well as for exchangeability of genes/domains and were excluded from any other unwanted position within the BGC.

A cloning plasmid with kanamycin resistance gene, p15A origin of replication, and restriction sites for assembly of the four DNA building blocks was constructed. Therefore, a multiple cloning site comprising *Dra*I, *Sda*I, *Apa*LI, *Nco*I, *Sal*I, *Acl*I, *Aat*II, *Ngo*MIV, *Avr*II, *Pac*I, and *Bam*HI restriction sites (0.1 kb) was amplified by

PCR using overlapping primers MCS for pACYC_fwd and MCS for pACYC_rev. PCR was performed with Phusion DNA polymerase (Thermo Scientific) under standard conditions according to the manufacturer's protocol. The reactions contained 5% DMSO and were carried out in an Eppendorf Mastercycler under the following conditions: initial denaturation for 2 min at 98 °C; 30 cycles consisting of denaturation for 15 s at 98 °C, annealing for 20 s at 72 °C, and extension for 10 s kb$^{-1}$ at 72 °C; and a final extension for 10 min at 72 °C. The MCS was inserted into plasmid pACYC177 via DraI and BamHI restriction sites, yielding plasmid pACYC_assembly.

The four building blocks of synthetic BGC version C1_V1 were stitched together in plasmid pACYC_assembly: building block 1 was inserted into pACYC_assembly via SdaI and ApaLI restriction sites, generating plasmid pACYC_BB1_C1_V1; building block 4 was inserted into pACYC_BB1_C1_V1 via AvrII and PacI restriction sites, generating plasmid pACYC_BB1+4_C1_V1; building block 2 was inserted into pACYC_BB1+4_C1_V1 via ApaLI and AclI restriction sites, generating plasmid pACYC_BB1+2+4_C1_V1; building block 3 was inserted into pACYC_BB1+2+4_C1_V1 via AclI and AvrII restriction sites, generating plasmid pACYC_BB1-4_C1_V1. The backbone of plasmid pACYC_BB1-4_C1_V1 was exchanged for the backbone of plasmid pINA1312-SdaI-cm$^R$-PacI via SdaI and PacI restriction sites, yielding plasmid pAf2. Plasmid pAf4 was constructed for targeted genome integration by exchange of the backbone of plasmid pAf2 for the backbone of plasmid pKG2-PIS via SdaI and PacI restriction sites.

The six building blocks of synthetic BGC version C3 were stitched together in plasmids pACYC_assembly and pAf2: building block 1 was inserted into pACYC_assembly via SdaI and ApaLI restriction sites, generating plasmid pACYC_BB1_C3; building block 2 was inserted into pACYC_BB1_C3 via ApaLI and NcoI restriction sites, generating plasmid pACYC_BB1+2_C3; building block 3 was inserted into pACYC_BB1+2_C3 via NcoI and AclI restriction sites, generating plasmid pACYC_BB1-3_C3; building blocks 1–3 of pACYC_BB1-3_C3 were inserted into pAf2 via SdaI and AclI restriction sites, generating plasmid pSyn_BB1-3_C3; building block 4 was inserted into pSyn_BB1-3_C3 via AclI and KpnI restriction sites, generating plasmid pSyn_BB1-4_C3; building block 5 was inserted into pSyn_BB1-4_C3 via KpnI and AvrII restriction sites, generating plasmid pSyn_BB1-5_C3; building block 6 was inserted into pSyn_BB1-5_C3 via AvrII and PacI restriction sites, generating plasmid pSyn_BB1-6_C3. The backbone of plasmid pSyn_BB1-6_C3 was exchanged for the backbone of plasmid pACYC_assembly or for the backbone of plasmid pKG2-PIS via SdaI and PacI restriction sites, yielding either plasmid pACYC_BB1-6_C3 or plasmid pMr1. Sequences of primers used in this study are listed in Supplementary Table 11 and further information on the plasmids constructed in this study are provided in Supplementary Table 12.

For the heterologous expression of the artificial PUFA biosynthetic pathway originating from *M. rosea* (SBNa008) with improved 5′ coding sequences in *Y. lipolytica*, the existing plasmid pACYC_BB1-6_C3 was modified in the following way: building block 6 was exchanged for BB6_C3_mod 5′ via AvrII and PacI restriction sites, generating plasmid pACYC_BB1-6_C3_BB6_mod 5′; the 5′ part of building block 2 of pACYC_BB1-6_C3_BB6_mod 5′ was exchanged for BB2_C3_mod 5′ via ApaLI and PciI restriction sites, generating plasmid pACYC_BB1-6_C3_BB2+6_mod 5′; the 5′ part of building block 4 of pACYC_BB1-6_C3_BB2+6_mod 5′ was exchanged for BB4_C3_mod 5′ via AclI and KpnI restriction sites, generating plasmid pACYC_BB1-6_C3_BB2+4+6_mod 5′; building block 1 of pACYC_BB1-6_C3_BB2+4+6_mod 5′ was exchanged for BB1_C3_mod 5′ via SdaI and ApaLI restriction sites, generating plasmid pACYC_BB1-6_C3_BB1+2+4+6_mod 5′ (Supplementary Table 12). The backbone of plasmid pACYC_BB1-6_C3_BB1+2+4+6_mod 5′ was exchanged for the backbone of plasmid pKG2-PIS via SdaI and PacI restriction sites, yielding plasmid pMr2 (Supplementary Table 12).

**Cloning of expression plasmids containing chimeric *pfa* BGCs.** For the heterologous expression of artificial hybrid PUFA biosynthetic pathways in *Y. lipolytica*, several chimeric PUFA expression constructs were generated. Plasmid pACYC_SynHyb1 was generated by replacement of hp4d, the KS2, the CLF, the AT2, the DH2, and the DH3 domain of synthetic gene *pfa3* originating from *A. fasciculatus* (SBSr002) located on plasmid pACYC_BB1-4_C1_V1 by the homologous domains of synthetic gene *pfa3* originating from *M. rosea* (SBNa008) located on plasmid pSyn_BB1-6_C3 via AclI and FseI restriction sites. The backbone of plasmid pACYC_SynHyb1 was exchanged for the backbone of plasmid pKG2-PIS via SdaI and PacI restriction sites, yielding plasmid pHyb2. The DH2, the DH3, and the AGPAT domain of synthetic chimeric gene *pfa3* located on plasmid pHyb2 were replaced by the homologous domains plus the DH4 domain of synthetic gene *pfa3* originating from *M. rosea* (SBNa008) located on plasmid pACYC_BB1-6_C3 via KpnI and AvrII restriction sites, generating plasmid pHyb1. Plasmid pHyb2a was generated by replacing the AGPAT domain of synthetic chimeric gene *pfa3* located on plasmid pHyb2 by a synthetic DNA fragment, consisting of the DH4 domain of synthetic gene *pfa3* from *M. rosea* (SBNa008) plus the AGPAT domain of synthetic gene *pfa3* originating from *A. fasciculatus* (SBSr002), via FseI and AvrII restriction sites. Plasmid pHyb5 was generated by replacing the DH2 and the DH3 domain of synthetic gene *pfa3* originating from *A. fasciculatus* (SBSr002) located on plasmid pAf4 by the homologous domains of synthetic gene *pfa3* originating from *M. rosea* (SBNa008) located on plasmid

pACYC_BB1-6_C3 via KpnI and FseI restriction sites. Plasmid pHyb5a was generated by replacing the AGPAT domain of synthetic gene *pfa3* located on plasmid pHyb5 by a synthetic DNA fragment, consisting of the DH4 domain of synthetic gene *pfa3* from *M. rosea* (SBNa008) plus the AGPAT domain of synthetic gene *pfa3* originating from *A. fasciculatus* (SBSr002), via FseI and AvrII restriction sites. The AGPAT domain of synthetic gene *pfa3* originating from *A. fasciculatus* (SBSr002) located on plasmid pAf4 was replaced by the DH4 domain plus the AGPAT domain of synthetic gene *pfa3* originating from *M. rosea* (SBNa008) located on plasmid pACYC_BB1-6_C3 via FseI and AvrII, yielding plasmid pHyb6. The AGPAT domain of synthetic gene *pfa3* located on plasmid pHyb6 was replaced by a synthetic DNA fragment consisting of the DH4 domain of synthetic gene *pfa3* from *M. rosea* (SBNa008) plus the AGPAT domain of synthetic gene *pfa3* originating from *A. fasciculatus* (SBSr002), via FseI and AvrII restriction sites, yielding plasmid pHyb6b. In order to inactivate the DH4 domain of plasmid pHyb6b by an active site H2270A point mutation, the DH4 and the AGPAT domain were amplified in two fragments from plasmid pHyb6b using primers H2270A_fwd_2/H2270A_CA to GC_rev and H2270A_CA to GC_fwd/H2270_rev. The first fragment (0.2 kb) carried the CA to GC nucleotide exchanges at the 3′ end, whereas the second fragment (2.2 kb) carried these base exchanges at the 5′ end. PCR amplification of the two fragments to be spliced was performed with Phusion DNA polymerase (Thermo Scientific) under standard conditions according to the manufacturer's protocol. The reactions contained 5% DMSO and were carried out in an Eppendorf Mastercycler under the following conditions: initial denaturation for 2 min at 98 °C; 20 cycles consisting of denaturation for 15 s at 98 °C, annealing for 20 s at 72 °C, and extension for 10 s kb$^{-1}$ at 72 °C; and a final extension for 10 min at 72 °C. For the subsequent overlap extension PCR using primers H2270A_fwd_2 and H2270A_rev, the two amplified fragments were used as templates. PCR of the 2.4 kb fragment was performed as described for the amplification of the two fragments to be spliced. In order to generate plasmid pHyb6b-H2270A, the DH4 and the AGPAT domain of plasmid pHyb6b were replaced by the PCR amplicon via FseI and AvrII restriction sites. Plasmid pHyb7 was generated by replacing synthetic genes *pfa2* and *pfa3* originating from *A. fasciculatus* (SBSr002) located on plasmid pAf4 by synthetic genes *pfa2* and *pfa3* originating from *M. rosea* (SBNa008) located on plasmid pACYC_BB1-6_C3 via ApaLI and AvrII restriction sites. Plasmid pACYC_SynHyb3 was generated by replacement of synthetic gene *pfa2* originating from *A. fasciculatus* (SBSr002) located on plasmid pACYC_BB1-4_C1_V1 by synthetic gene *pfa2* originating from *M. rosea* (SBNa008) located on plasmid pMr1 via ApaLI and AclI restriction sites. The backbone of plasmid pACYC_SynHyb3 was exchanged for the backbone of plasmid pKG2-PIS via SdaI and PacI restriction sites, yielding plasmid pHyb8. The DH1 domain of synthetic gene *pfa2* originating from *M. rosea* (SBNa008) (1.2 kb) was amplified from plasmid pACYC_BB1-6_C3 using primers DH1-Mr_KflI_fwd and linker 2_rev. PCR was performed with Phusion DNA polymerase (Thermo Scientific) under standard conditions according to the manufacturer's protocol. The reactions contained 5% DMSO and were carried out in an Eppendorf Mastercycler under the following conditions: initial denaturation for 2 min at 98 °C; 20 cycles consisting of denaturation for 15 s at 98 °C, annealing for 20 s at 66 °C, and extension for 10 s kb$^{-1}$ at 72 °C; and a final extension for 10 min at 72 °C. In order to generate plasmid pACYC_SynHyb4, the DH1 domain of synthetic gene *pfa2* originating from *A. fasciculatus* (SBSr002) located on plasmid pACYC_BB1-4_C1_V1 was replaced by the PCR amplicon via KflI and AclI restriction sites. The backbone of plasmid pACYC_SynHyb4 was exchanged for the backbone of plasmid pKG2-PIS via SdaI and PacI restriction sites, yielding plasmid pHyb9. For the fusion of the KR domain of synthetic gene *pfa2* originating from *M. rosea* (SBNa008) and the DH1 domain of synthetic gene *pfa2* originating from *A. fasciculatus* (SBSr002), the KR domain was amplified as the first fragment (1.2 kb) from plasmid pMr2 using primers KR-Mr_fwd and KR-Mr+DH1-Af overlap_rev. The second fragment (1.1 kb) contained the DH1 domain and was amplified from plasmid pAf4 using primers DH1-Af+KR-Mr overlap_fwd and linker 2_rev. PCR amplification of the two fragments to be spliced was performed with Phusion DNA polymerase (Thermo Scientific) under standard conditions according to the manufacturer's protocol. The reactions contained 5% DMSO and were carried out in an Eppendorf Mastercycler under the following conditions: initial denaturation for 2 min at 98 °C; 20 cycles consisting of denaturation for 15 s at 98 °C, annealing for 20 s at 67 °C, and extension for 10 s kb$^{-1}$ at 72 °C; and a final extension for 10 min at 72 °C. For the subsequent overlap extension PCR using primers KR-Mr_fwd and linker 2_rev, the two amplified fragments were used as templates. PCR of the 2.2 kb fragment was performed as described for the amplification of the two fragments to be spliced. In order to generate plasmid pACYC_SynHyb7, the KR and the DH1 domain of synthetic gene *pfa2* originating from *M. rosea* (SBNa008) located on plasmid pACYC_BB1-6_C3_BB1+2+4+6_mod 5′ was replaced by the PCR amplicon via SalI and AclI restriction sites. The backbone of plasmid pACYC_SynHyb7 was exchanged for the backbone of plasmid pKG2-PIS via SdaI and PacI restriction sites, yielding plasmid pHyb15. Sequences of primers used in this study are listed in Supplementary Table 11 and further information on the plasmids constructed in this study are provided in Supplementary Table 12.

**Standard LC-PUFA production.** For heterologous LC-PUFA production in *Y. lipolytica*, cultivations were carried out in 10 mL YNBG medium in 50 mL shake flasks. The medium was inoculated with an overnight culture (1:100) and incubated

at 28 °C for 168 h. Afterwards, 1 mL of the culture was subjected to FAME extraction (see below).

**Medium optimization for enhanced LC-PUFA production.** In a number of studies, the composition of the YNBG production medium was varied to investigate the impact of different nutrients on LC-PUFA production. The experiments included (i) a variation of the primary carbon source (glucose, glycerol, acetate), (ii) a variation of the nitrogen source (NH₄Cl, (NH₄)₂SO₄), (iii) a variation of the C/N ratio in the range of 4–55), and (iv) a variation of phosphate level by an adjustment of the YNB concentration (0.85–1.7 g L⁻¹), and a replacement of potassium phosphate buffer by MES in a range of 50–200 mM. Cultures were grown at 28 °C in 500 mL baffled shake flasks with 50 mL medium on an orbital shaker (230 rpm).

**DHA production in a fed-batch process.** The production performance of the DHA producer *Y. lipolytica* Po1h::Af4 was evaluated in a fed-batch process. Fermentation was carried out in glucose and in glycerol-based medium using 1 L DASGIP bioreactors (Eppendorf, Jülich, Germany). The initial batch medium (300 mL) contained per liter: 25 g glucose or 25.6 g glycerol, 5 g (NH₄)₂SO₄ (C/N ratio 11), 1.7 g YNB w/o amino acids, 1 g KH₂PO₄, 200 mmol MES (pH 5.5), and 1 mL antifoam (Antifoam 204, Sigma, Germany). The process was inoculated with exponentially growing cells from an overnight pre-culture. Feeding (600 g L⁻¹ glucose or 600 g L⁻¹ glycerol) was started at a rate of 1.5 mL h⁻¹, when the substrate was depleted. During the feed phase, the level of the carbon source was monitored at-line. The data were used to re-adjust the feed rate in order to avoid a limitation of the carbon source. Cultivation temperature was maintained constant at 28 °C. The pH and the pO₂ level were monitored online with a pH electrode (Mettler Toledo, Gießen, Germany) and a pO₂ electrode (Hamilton, Höchst, Germany). The pH was kept constant at 5.5 ± 0.05 by automated addition of 6 M NaOH and 6 M HCl. The dissolved oxygen level was maintained above 30% of saturation during the batch phase and was reduced to 5% during the feed phase by variation of stirrer speed and aeration rate. Data acquisition and process operations were controlled by the DASGIP control software 4.0 (Eppendorf, Jülich, Germany).

**Quantification of extracellular metabolites.** Glucose in culture supernatant was determined with a biochemical analyzer (YSI 2900, Kreienbaum, Germany). Sugar alcohols and organic acids were quantified by high-performance liquid chromatography (Agilent 1200 series, Waldbronn, Germany) equipped with a ultraviolet and refraction index detector using an Aminex HPX-87H column (Bio-Rad, Hercules, USA) at 45 °C and isocratic elution with 12 mM H₂SO₄ at a flow rate of 0.5 mL min⁻¹. Ammonium was quantified using a photometric kit (Spectroquant Ammonium Test, Merck, Darmstadt, Germany).

**Preparation of FAMEs.** The cells were harvested by centrifugation at 12,000 × g for 5 min, transferred to a glass vial, and dried in a vacuum concentrator. Subsequently, the CDW was determined. The cellular fatty acids were extracted using the FAME method[31]. For this purpose, the cells were transferred to a glass vial and dried in a vacuum concentrator. In all, 25–50 µg of n-3 heneicosapentaenoic acid (HPA; 22:5) methyl ester (Cayman Chemical) or n-3 DPA methyl ester (Sigma-Aldrich) and 250–500 µL of a mixture of methanol, toluene, and sulfuric acid (50:50:2, v/v/v) were added. The vial was capped with a Teflon-lined screw cap and incubated at 80 °C overnight. After the mixture was cooled to room temperature, 200–400 µL of an aqueous solution consisting of 0.5 M NH₄HCO₃ and 2 M KCl were added, and the sample was vortexed for 30 s. Phase separation was achieved by centrifugation at 4000 × g for 5 min at room temperature. Seventy-five microliters of the upper phase were mixed with 25 µL N-methyl-N-(trimethylsilyl)trifluoroacetamide and incubated at 37 °C for 30 min. Subsequently, the sample was used for GC-MS analysis.

**Extraction and fractionation of lipids.** Extraction of lipids from microbial cells was carried out using the method of Bligh and Dyer[32], modified by Lewis et al.[33], on a small scale. In the first step, the cell pellet from a 15 mL culture was transferred into a polypropylene tube. Successively, 4 mL chloroform, 8 mL methanol, and 3.2 mL of 1% NaCl were added, and the tube was vortexed at high speed for 15 s after every addition. The sample was agitated on a tube rotator at 30 rpm overnight. Four milliliters of chloroform and 4 mL of 1% NaCl were then added, and the tube was inverted 30 times. Phase separation was achieved by centrifugation at 4000 × g for 5 min at room temperature. The bottom layer containing the lipid extract was evaporated to dryness under a gentle stream of nitrogen and dissolved in 1 mL of a chloroform:methanol mixture (2:1, v/v). The lipid solution was fractionated by solid-phase extraction on a Strata SI-1 silica column with 100 mg sorbent mass (Phenomenex) equilibrated with chloroform+1% acetic acid. Neutral lipids and free fatty acids were eluted from the column with 1 mL of chloroform+1% acetic acid, glycolipids were eluted with 1.5 mL of an acetone: methanol mixture (9:1, v/v), and phospholipids were eluted with 1 mL methanol. For GC-MS analysis, the fractions were dried in a vacuum concentrator and further processed according to the FAME method.

**Analysis of FAMEs by GC-MS.** GC-MS was carried out on an Agilent 6890N gas chromatograph (Agilent Technologies) equipped with a 7683B split/splitless injector with autosampler (Agilent Technologies) and coupled to a 5973 electron impact mass-selective detector (Agilent Technologies). Helium was used as the carrier gas at a flow rate of 1 mL min⁻¹. A measure of 0.2–5 µL of the sample were injected in split mode (split ratio, 10:1). The analytical column was a (5% phenyl)-methylpolysiloxane capillary column (Agilent J&W DB-5ht; 30 m × 0.25 mm i.d. × 0.1 µm film thickness, maximum temperature 400 °C; Agilent Technologies). The column temperature was kept at 130 °C for 2.5 min, increased to 240 °C at a rate of 5 °C min⁻¹, then ramped to 300 °C at 30 °C min⁻¹, and held at 300 °C for 5 min. Other temperatures were as follows: inlet, 275 °C; GC-MS transfer line, 280 °C; ion source, 230 °C; and quadrupole, 150 °C. The mass-selective detector was operated in scan mode, scanning the mass range from m/z 40 to 700. Scan control, data acquisition, and processing were performed by MSD ChemStation and AMDIS software, version 2.69, based on the masses, fragmentation patterns, and retention times, in comparison with Supelco 37 Component FAME Mix and LC-PUFA methyl esters (all Sigma-Aldrich) as reference standards, and NIST 08 database. Absolute amounts of PUFAs were quantified by integration of the peaks using MSD ChemStation and by subsequent calculation in relation to the integral of n-3 HPA methyl ester or n-3 DPA methyl ester and to CDW.

**Reporting summary.** Further information on research design is available in the Nature Research Reporting Summary linked to this article.

## Data availability

The sequences of the synthetic PUFA BGCs version C1_V1, C1_V2, C3, and C3_mod 5′ have been deposited in the GenBank database (accession numbers MN047805, MN047806, MN047807, and MN047808). The source data underlying Figs. 2, 3, 4, and 5, Supplementary Figs. 12 and 19, and Supplementary Tables 9 and 10 are provided as a Source Data file. All other relevant data are available from the authors upon reasonable request.

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

## Acknowledgements

We thank Nestor Zaburannyi for analyses of Illumina sequence data, Catherine Madzak for plasmid pINA1312, and Isabelle Kocker for support during medium development. This work was generously supported by a grant from the European Union (grant no. C/4-EFRE-11/2008), by a grant from the Federal Ministry of Economic Affairs and Energy (grant no. 03SHWB065), and by grants from the Federal Ministry of Education and Research (grant nos. 031B0346A and 031B0346B).

## Author contributions

K.G. performed all cloning experiments in the laboratory and analyzed the GC-MS data from strain evaluation. K.G, G.Z., H.S.B. and S.C.W. designed the artificial DNA sequences. G.Z. and H.S.B. performed the codon adaptation. D.D. and M.K. conducted medium development and fed-batch production of DHA, including all analytics. K.G., C.W., S.C.W. and R.M. designed the study. K.G., D.D., M.K., G.Z., H.S.B., C.W., S.C.W. and R.M. wrote the manuscript.

## Additional information

**Competing interests:** The authors declare no competing interests.

