## [Peer Review File · Nature Communications]

Reviewers' comments:

Reviewer #1 (Remarks to the Author):

This manuscript reports *Yarrowia lipolytica* strains engineered to produce LG-PUFAs, which are valuable due to their health benefits. There have been several previous reports on *Yarrowia* strains engineered to produce these products, which the authors cite. This study uses a different approach to making LG-PUFAs, employing polyketide synthase-like PUFA synthases instead of fatty acid biosynthesis and desaturases. However, The DPA/DHA-type pfa BGC has been engineered in other heterologous hosts before. Therefore, while this is the first report of engineering BGC in *Y. lipolytica*, previous studies of PUFA production in *Yarrowia*, and BGC in other organisms take away some of the novelty from this study in this regard.

Nevertheless, engineering *Yarrowia* to produce LG-PUFAs has several advantages that the authors discuss. The authors do codon optimization and harmonization, but this does not seem to be very effective. The authors also build chimeras that change the product profiles (which is not very surprising), but it is not clear what mechanistic insights come from these experiments.

The writing quality is acceptable but the long names used for their strains makes it difficult to follow the text and figures at times. In addition, the authors' priorities for what they choose to write in the Results section are puzzling and take away from their study. Much of the Results section reads like the plasmid construction section in what is usually in the Methods section at the expense of writing more useful explanations on their codon optimization/harmonization techniques or chimera designs.

The main text could benefit from a better explanation of their "codon harmonization" strategy. It is not sufficient to just report what they did. They should also explain the concept. It is ambiguous when they say they "replace a less well-adapted codon with a rarer one" and how this could benefit gene expression. The supplementary note 1 is not very helpful. It is fine to keep the details of their sequence designs in a supplementary note, but this should not be a substitute for describing, at least in general terms what the strategy for the design is. Instead they present in the results a lot of details on vector construction (which is better suited for the methods) and no description of their sequence design strategy, or the strategies followed in their chimera designs.

Nevertheless, the almost identical levels of pfa2-mcherry production using different codons raises the question of the effectiveness of their codon harmonization technique. The alternative explanation they give in page 10 is not convincing.

They need to provide the final sequences for the three pfa genes and ppt they used to see what are the codon usage in each experiment

Yarrowia can't grow anaerobically, so what do they mean that the "(PKS)-like PUFA synthase from myxobacteria enable anaerobic de novo LC-PUFA biosynthesis"?

It's not clear what the white and gray shades in figures 2a and 2b are meant to represent.

Their statement that YALIO_C05907g has emerged as a good integration site for gene expression (page 8) needs a reference.

The color code in Fig 2c is hard to interpret. Furthermore, they reference to this figure when discussing DHA production is strain Polh::SynPFaPptAf2 clone C in page 8, but this clone does not seem to be shown in this figure. It is very confusing.

Not clear what they mean by "energetic and structural mRNA sequence calculations for genes pfa1, pfa2, pfa3, and ppt of" in page 11.

To better understand the strategy to designing their BGC chimeras and the effects they produce, it would be very helpful to show in a figure (maybe a modified Figure 1) what the catalytic activities of Pfa1, Pfa2, Pfa3 and Ppt are.

The DNA or amino acid sequence of their chimeras should be reported to specify cut lengths and linkers.

The BGC chimeras they design shift the products that they make, which is interesting. However their descriptions are difficult to follow. It would be helpful to make a figure to follow their strategy, summarize the main conclusion they draw and the lessons they learned about how the different domains of these BGCs influence product specificity.

It is clear that the combination of multiple functional domains determines the length and number/position of double bonds in the products, but do they know how this works? What mechanistic insights does this study provide? The level of understanding the authors attain will determine the true impact of this study. Unfortunately, the level of understanding of this question does not come across in the current manuscript.

In conclusion, this reviewer is not convinced that the novelty and impact of this study rises to the level of the average publication in Nature Communications.

Reviewer #2 (Remarks to the Author):

Gemperlein et al. describe the rational engineering of biosynthetic gene clusters (BGCs) for the production of polyunsaturated fatty acids (PUFAs). PUFAs contain more than one double bond in the unbranched alkyl chain. Some PUFAs belong to the class of "essential fatty acids". PUFAs are particularly interesting, because they confer positive effects on a set of diseases, including heart diseases, cancers, inflammations, and diabetes. Any sustainable access to these compounds is valuable in order to meet the rising demands for these compounds. Gemperlein et al. select *Yarrowia lipolytica* as production host for PUFAs, because this strain is able to produce high amounts of lipids.

PKSs work iteratively by assembling C₂ units to the desired compound. Myxobacterial PUFA BGCs are composed of two PKSs that may work in sequence. The authors aim at producing PUFAs in *Y. lipolytica* by inserting PKS encoding sequences, known to produce PUFAs in myxobacteria, in the *Y. lipolytica* genome. This approach is followed with a set of thoroughly executed experiments that are presented in the manuscript in logical manner. Authors eventually present *Y. lipolytica* strains, which allow producing high titers of selected long chain length PUFAs (LC-PUFAs). PUFAs are analyzed as FAMES after fatty acid extraction from *Y. lipolytica* cultures. Interestingly, PUFA BGCs from *A. fasciculatus* and from *M. rosae* produce different products in *Y. lipolytica*, with respect to chain length, position of double bonds and number of double bonds, although these BGCs are highly similar. Chimeric

constructs, swapping larger parts of the BGCs as well as single domains, shift the product spectra of PUFAs.

The presented approach is superior to a previously represented attempt for PUFA synthesis (Xue et al. 2016 Nat Biotechnol) in which separate proteins (desaturases and elongases) were inserted in *Y. lipolytica* in order to redirect fatty acids to unsaturation. The advancement in the approach by Gemperlein et al. lies in harnessing the compartmentalized synthesis scheme of the PUFA-producing PKSs, which allows improved product control, and independence from the complex fatty acid metabolism.

General point:

(1) Although the value of the final strains is undisputed, the study itself is moderately novel. The approach is based on earlier work in which the BGCs have been harnessed for PUFA synthesis already. In 2016, the authors have reported the production of docosahexaenoic acid in *Pseudomonas putida*, although in lower yields. In this new study, the oleaginous yeast *Y. lipolytica* was taken as a production host. The choice of *Y. lipolytica* is not surprising, when considering the previous different approach for PUFA synthesis in this organism (Xue et al. 2016 Nat Biotechnol).

Innovation is in some details of this study: (a) Codon harmonization as a tool to improve protein quality and yield is surely innovative in BGC design. Authors test the effect of codon harmonization in product yield and by in-cell fluorescence originating from C-terminally attached FP. Product yields and spectra are not affected by codon harmonization. Codon harmonization is meant to influence the quality of protein; i.e. proper folding as partly indicated by soluble protein, than to act on overall protein concentration. Monitoring the fluorescence of a C-terminal FP does therefore not necessarily indicate the effect of harmonization unless the FP acts as a reporter of protein quality. Is this so? (see ref. Waldo et al. 2009 Nat Biotechnol)

(b) The construction chimeric BGCs turns out to be powerful. Although the molecular origin of effects remains vague (see below), the respective strains are valuable for production of distinct PUFAs.

(2) The authors claim to dissect “the molecular basis of for the specificity of PUFA synthase-catalyzed reactions” (line 275). They swap parts of the BGCs to produce chimeric PUFA-PKSs and domains, and analyze the product spectrum. I do not see that the molecular basis of the PUFA synthesis is indeed revealed by these experiments, and in several cases I cannot follow the conclusions drawn by the authors; in the following highlighted on the example of the impact of DH domains for the spectrum of produced PUFA (lines 281-303). In a series of three constructs, shown in Fig. 3a, the swapped region is reduced from the entire *pfa3* to just the DH domains. Since the EPA and n-3 DPA are produced as major products, ignoring that there are severe shifts in the EPA vs. n-3 DPA product ratio (Ppt2a vs. Ppt5a), the authors conclude that the DH domains determine the product specificity of *pka3*. This is speculation, mainly because the knowledge about the synthesis is poor for such precise statements. For example, it is unclear which module (*pka2* or 3) contributes to which part of the product(s). Further the kinetics of the synthesis are not understood, so that the impact of even small changes of the biosynthesis, for example by introducing non-native domain-domain interactions in *pka3* (ACP:DHs), remains unclear.

The argument “DH domains defining products” is too simple, as shown by other data in the manuscript (see data in Figure 3d and constructs in Figure 3c). Construct Ppt6b can be seen as a part of construct series Ppt1 – Ppt2a – Ppt5a; i.e., compared to Ppt5a it is further reduced in the swapped region to a simple DH4 exchange. In this construct (Ppt6b), the spectrum is severely changed. A further construct with a functional knockout of DH4, construct Ppt6-H2270A, restores the spectrum of the wildtype-like construct Af4. Data on Ppt6b and Ppt6-H2270A is remarkable in two aspects: (i) Construct Ppt6b shows how strongly just the swapped DH4 affects the substrate spectrum, because it overwrites the cognate DH (DH, DH2 and DH3). (ii) Construct Ppt6-H2270A shows that the cognate DH4 does not

seem to be important for the products spectrum, because the PPT6-H2270A produces as similar spectrum as Af4. Both data shows that each of the DH domain likely has its own complex influence on the product spectrum.

Enzyme kinetics of multi-domain proteins is extremely complex, and product spectra are therefore just partially suited read-outs for analyzing the molecular basis of such proteins (see for example Gajewski et al. 2017 Nat Chem Biol). Overall, the experimental set-up of Gemperlein et al. in swapping BGC parts and domains, thereby interfering in structure, domain-domain interactions and specificity, is too complex for specific statements to the molecular basis of PUFA synthesis. A better experimental set-up for characterizing PUFA synthesis in detail better involves functional knockouts of domains, or domain depletion and duplications, which has been performed for such systems before (see for example Hayashi et al. 2016 Sci Rep, for analyzing the impact of ACP domains). I recommend refraining from extracting any deeper information from this dataset, but rather stressing the technological relevance of shifting product spectra by chimeric BGCs.

Minor points:

(1) Although the approach is clear and experiments thoroughly described, the improved figures and a different arrangement of figure panels would make the manuscript better readable. For example, Figure 2a shows the BGC of *A. fasciculatus*. PUFA data to this BGC in *Y. lipolytica* is shown in Figure 2c together with data from the second BGC (*Minicystis rosea*). The *Minicystis rosea* BGC is, however, introduced much later. It is very difficult to read data in Figure 2c when the information to the second BGC is withheld several pages. Please rearrange this chapter or the figure panels. In addition, the squares for showing the color code are too small and colors are too similar. Figure legend also misses label "c".

I also recommend showing structures of the main polyunsaturated fatty acids produced by myxobacteria, and recommend improving Figure 1 by including AGPAT in the reaction scheme. Any information that is available on the interplay of pfa1, 2 and 3 should be included as well.

(2) The nomenclature of the strains/plasmid is very difficult. It needed the printed figures next to the screen to be able to decode the results section and to connect data to constructs. Please think of a simpler nomenclature? There are 11 different clusters shown. It may be possible to assign a specific color to each of this cluster, and than color code data presentation. All this would make the paper better readable.

(3) For comparison, include data of wildtype-like BGCs (shown in Figure 2) in Figure 3.

(6) Can parts be omitted from the main text? All data to codon harmonization are presented in the SI. Also the chapter to codon harmonization could be moved from the main text to SI. Details to construct Ppt7 could be moved to SI, too. A sentence in the main text saying that ER exchanges do not influence the product spectrum should be enough.

The work of Gemperlein et al. surely warrants publication. Although the approach is not highly novel (see Gemperlein 2014 Chem Sci, and Gemperlein et al. 2016 Metabolic Engineering), the study demonstrates the successful production of PUFAs at yet unmatched titers and variability. The suitability of *Y. lipolytica* for production of PUFAs, a thorough experimental set-up as well as the huge experimental effort makes this study a success, which is worth sharing with the readership of Nature Communications.

We thank the reviewers for their critical and helpful comments, which we have addressed in the revised manuscript. Given below is a point-by-point response to each of the reviewers' comments.

Reviewer #1 (Remarks to the Author):

This manuscript reports *Yarrowia lipolytica* strains engineered to produce LG-PUFAs, which are valuable due to their health benefits. There have been several previous reports on *Yarrowia* strains engineered to produce these products, which the authors cite. This study uses a different approach to making LG-PUFAs, employing polyketide synthase-like PUFA synthases instead of fatty acid biosynthesis and desaturases. However, The DPA/DHA-type pfa BGC has been engineered in other heterologous hosts before. Therefore, while this is the first report of engineering BGC in *Y. lipolytica*, previous studies of PUFA production in *Yarrowia*, and BGC in other organisms take away some of the novelty from this study in this regard.

Nevertheless, engineering *Yarrowia* to produce LG-PUFAs has several advantages that the authors discuss. The authors do codon optimization and harmonization, but this does not seem to be very effective. The authors also build chimeras that change the product profiles (which is not very surprising), but it is not clear what mechanistic insights come from these experiments.

The writing quality is acceptable but the long names used for their strains makes it difficult to follow the text and figures at times. In addition, the authors' priorities for what they choose to write in the Results section are puzzling and take away from their study. Much of the Results section reads like the plasmid construction section in what is usually in the Methods section at the expense of writing more useful explanations on their codon optimization/harmonization techniques or chimera designs.

The long names of plasmids and strains were shortened according to the reviewer's suggestion. Additionally, the description of the plasmid constructions was significantly reduced in the "Results" section and the explanations of the "codon harmonization" strategy and the chimera design were extended accordingly.

The main text could benefit from a better explanation of their "codon harmonization" strategy. It is not sufficient to just report what they did. They should also explain the concept. It is ambiguous when they say they "replace a less well-adapted codon with a rarer one" and how this could benefit gene expression. The supplementary note 1 is not very helpful. It is fine to keep the details of their sequence designs in a supplementary note, but this should not be a substitute for describing, at least in general terms what the strategy for the design is. Instead they present in the results a lot of details

on vector construction (which is better suited for the methods) and no description of their sequence design strategy, or the strategies followed in their chimera designs. Nevertheless, the almost identical levels of *pfa2*-mcherry production using different codons raises the question of the effectiveness of their codon harmonization technique. The alternative explanation they give in page 10 is not convincing.

We extended the introduction to the "codon harmonization" strategy as follows: "Thereby, the transfer of the course of the codon usage along the message from the prokaryotic donor to the eukaryotic acceptor was regarded as the simplest formalism of codon adaptation with a high plausibility." We extended the description of the "codon harmonization" strategy with an explanation of the concept as follows: "Especially, rare codons should be conserved, as it has been shown that they have an important role for the production of functional proteins, potentially in the regulation of the rate of protein synthesis and of the earliest steps of protein folding." As our codon harmonization approach was indeed not very effective in terms of yield we decided to move the corresponding chapter "Design of a new version of an artificial DPA/DHA-type *pfa* gene cluster and heterologous LC-PUFA production" and the corresponding chapters into the "Methods" section from the main text to the SI.

They need to provide the final sequences for the three *pfa* genes and *ppt* they used to see what are the codon usage in each experiment

The sequences of the synthetic coding sequences of *pfa1*, *pfa2*, *pfa3*, and *ppt* originating from *Aerobacter fasciculatus* and *Minicystis rosea* have been deposited in the GenBank database (accession numbers MN047805, MN047806, MN047807, and MN047808).

Yarrowia can't grow anaerobically, so what do they mean that the "(PKS)-like PUFA synthase from myxobacteria enable anaerobic de novo LC-PUFA biosynthesis"?

In the context of biosynthesis, the term "anaerobic" LC-PUFA biosynthesis is related to the oxygen-independent biosynthesis of LC-PUFAs catalyzed by PUFA synthases. In contrast, the PUFA biosynthesis catalyzed by oxygen-dependent fatty acid desaturases is known as "aerobic" PUFA biosynthesis. Both terms are well-established in the field of fatty acid research and the difference between the two makes an important aspect of the paper which is described in detail in the text. However, to avoid any confusion, the word "anaerobic" was deleted in the text.

It's not clear what the white and gray shades in figures 2a and 2b are meant to represent.

Myxobacterial PUFA synthases contain multifunctional proteins. The different catalytic domains of the genes/proteins were shown as grey boxes. For a better discrimination of all the different PUFA gene clusters, the Figures 2 and 3 were transformed from grayscale figures into color figures.

Their statement that YALI0_C05907g has emerged as a good integration site for gene expression (page 8) needs a reference.

The finding that YALI0_C05907g serves as a good integration site was made during our study. To clarify this point, we extended the respective statement as follows: "In the course of the present study, locus YALI0_C05907g has emerged as an integration site that enables a good expression of recombinant *pfa* BGCs."

The color code in Fig 2c is hard to interpret. Furthermore, they reference to this figure when discussing DHA production is strain Polh::SynPFaPptAf2 clone C in page 8, but this clone does not seem to be shown in this figure. It is very confusing.

For a better discrimination of all the different PUFA-producing *Y. lipolytica* strains, Figures 2 and 3 were transformed from grayscale figures into color figures. The PUFA production profile of strain Po1h::SynPFaPptAf2 clone C is now also included in Fig. 2c.

Not clear what they mean by "energetic and structural mRNA sequence calculations for genes *pfa1*, *pfa2*, *pfa3*, and *ppt* of" in page 11.

The redesign of the 5' coding regions of all four genes was described more precisely as follows: "Calculations of the opening energies within the translation initiation sites of genes *pfa1*, *pfa2*, *pfa3*, and *ppt* of cluster C3 revealed the potential for improvement of the ribosomal access to the translational initiation region on mRNA level."

To better understand the strategy to designing their BGC chimeras and the effects they produce, it would be very helpful to show in a figure (maybe a modified Figure 1) what the catalytic activities of Pfa1, Pfa2, Pfa3 and Ppt are.

Fig. 1 and in particular the corresponding legend were adapted accordingly. The catalytic domains are now correlated with the respective Pfa protein.

The DNA or amino acid sequence of their chimeras should be reported to specify cut lengths and linkers.

The sequences of the synthetic coding sequences of *pfa1*, *pfa2*, *pfa3*, and *ppt* originating from *Aetherobacter fasciculatus* and *Minicystis rosea* have been deposited in the GenBank database

(accession numbers MN047805, MN047806, MN047807, and MN047808). Together with the detailed description of the construction of the chimeric sequences in the “Methods” section, all the sequences can easily be tracked and examined.

The BGC chimeras they design shift the products that they make, which is interesting. However their descriptions are difficult to follow. It would be helpful to make a figure to follow their strategy, summarize the main conclusion they draw and the lessons they learned about how the different domains of these BGCs influence product specificity. It is clear that the combination of multiple functional domains determines the length and number/position of double bonds in the products, but do they know how this works? What mechanistic insights does this study provide? The level of understanding the authors attain will determine the true impact of this study. Unfortunately, the level of understanding of this question does not come across in the current manuscript. In conclusion, this reviewer is not convinced that the novelty and impact of this study rises to the level of the average publication in Nature Communications.

We believe that the changes made as described above and below clarify most of the points raised by reviewer #1. However, we have to admit that we and the whole research field are still far from a complete mechanistic understanding of multimodular and highly complex PUFA synthases (see also comments of reviewer #2, especially regarding the importance of the biotechnological relevance of our manuscript). Investigation of the biosynthesis, however, is not the main goal of this work. The overall impact of the current study was judged very favorably by reviewer #2.

Reviewer #2 (Remarks to the Author):

Gemperlein et al. describe the rational engineering of biosynthetic gene clusters (BGCs) for the production of polyunsaturated fatty acids (PUFAs). PUFAs contain more than one double bond in the unbranched alkyl chain. Some PUFAs belong the class of “essential fatty acids”. PUFAs are particularly interesting, because they confer positive effects on a set of diseases, including heart diseases, cancers, inflammations, and diabetes. Any sustainable access to these compounds is valuable in order to meet the raising demands for these compounds. Gemperlein et al. select *Yarrowia lipolytica* as production host for PUFAs, because this strain is able to produce high amounts of lipids. PKSs work iteratively by assembling C2 units to the desired compound. Myxobacterial PUFA BGCs are composed of two PKSs that may work in sequence. The authors aim at producing PUFAs in *Y. lipolytica* by inserting PKS encoding sequences, known to produce PUFAs in myxobacteria, in the *Y. lipolytica* genome. This approach is followed with a set of thoroughly executed experiments that are

presented in the manuscript in logical manner. Authors eventually present *Y. lipolytica* strains, which allow producing high titers of selected long chain length PUFAs (LC-PUFAs). PUFAs are analyzed as FAMES after fatty acid extraction from *Y. lipolytica* cultures. Interestingly, PUFA BGCs from *A. fasciculatus* and from *M. rosae* produce different products in *Y. lipolytica*, with respect to chain length, position of double bonds and number of double bonds, although these BGCs are highly similar. Chimeric constructs, swapping larger parts of the BGCs as well as single domains, shift the product spectra of PUFAs.

The presented approach is superior to a previously represented attempt for PUFA synthesis (Xue et al. 2016 Nat Biotechnol) in which separate proteins (desaturases and elongases) were inserted in *Y. lipolytica* in order to redirect fatty acids to unsaturation. The advancement in the approach by Gemperlein et al. lies in harnessing the compartmentalized synthesis scheme of the PUFA-producing PKSs, which allows improved product control, and independence from the complex fatty acid metabolism.

General point:

(1) Although the value of the final strains is undisputed, the study itself is moderately novel. The approach is based on earlier work in which the BGCs have been harnessed for PUFA synthesis already. In 2016, the authors have reported the production of docosahexaenoic acid in *Pseudomonas putida*, although in lower yields. In this new study, the oleaginous yeast *Y. lipolytica* was taken as a production host. The choice of *Y. lipolytica* is not surprising, when considering the previous different approach for PUFA synthesis in this organism (Xue et al. 2016 Nat Biotechnol). Innovation is in some details of this study: (a) Codon harmonization as a tool to improve protein quality and yield is surely innovative in BGC design. Authors test the effect of codon harmonization in product yield and by in-cell fluorescence originating from C-terminally attached FP. Product yields and spectra are not affected by codon harmonization. Codon harmonization is meant to influence the quality of protein; i.e. proper folding as partly indicated by soluble protein, than to act on overall protein concentration. Monitoring the fluorescence of a C-terminal FP does therefore not necessarily indicate the effect of harmonization unless the FP acts as a reporter of protein quality. Is this so? (see ref. Waldo et al. 2009 Nat Biotechnol) (b) The construction chimeric BGCs turns out to be powerful. Although the molecular origin of effects remains vague (see below), the respective strains are valuable for production of distinct PUFAs.

(2) The authors claim to dissect “the molecular basis of for the specificity of PUFA synthase-catalyzed reactions” (line 275). They swap parts of the BGCs to produce chimeric PUFA-PKSs and domains, and analyze the product spectrum. I do not see that the molecular basis of the PUFA synthesis is indeed

revealed by these experiments, and in several cases I cannot follow the conclusions drawn by the authors; in the following highlighted on the example of the impact of DH domains for the spectrum of produced PUFA (lines 281-303). In a series of three constructs, shown in Fig. 3a, the swapped region is reduced from the entire pfa3 to just the DH domains. Since the EPA and n-3 DPA are produced as major products, ignoring that there are severe shifts in the EPA vs. n-3 DPA product ratio (Ppt2a vs. Ppt5a), the authors conclude that the DH domains determine the product specificity of pka3. This is speculation, mainly because the knowledge about the synthesis is poor for such precise statements. For example, it is unclear which module (pka2 or 3) contributes to which part of the product(s). Further the kinetics of the synthesis are not understood, so that the impact of even small changes of the biosynthesis, for example by introducing non-native domain-domain interactions in pka3 (ACP:DHs), remains unclear. The argument “DH domains defining products” is too simple, as shown by other data in the manuscript (see data in Figure 3d and constructs in Figure 3c). Construct Ppt6b can be seen as a part of construct series Ppt1 – Ppt2a – Ppt5a; i.e., compared to Ppt5a it is further reduced in the swapped region to a simple DH4 exchange. In this construct (Ppt6b), the spectrum is severely changed. A further construct with a functional knockout of DH4, construct Ppt6-H2270A, restores the spectrum of the wildtype-like construct Af4. Data on Ppt6b and Ppt6-H2270A is remarkable in two aspects: (i) Construct Ppt6b shows how strongly just the swapped DH4 affects the substrate spectrum, because it overwrites the cognate DH (DH, DH2 and DH3). (ii) Construct Ppt6-H2270A shows that the cognate DH4 does not seem to be important for the products spectrum, because the Ppt6-H2270A produces as similar spectrum as Af4. Both data shows that each of the DH domain likely has its own complex influence on the product spectrum. Enzyme kinetics of multi-domain proteins is extremely complex, and product spectra are therefore just partially suited read-outs for analyzing the molecular basis of such proteins (see for example Gajewski et al. 2017 Nat Chem Biol). Overall, the experimental set-up of Gemperlein et al. in swapping BGC parts and domains, thereby interfering in structure, domain-domain interactions and specificity, is too complex for specific statements to the molecular basis of PUFA synthesis. A better experimental set-up for characterizing PUFA synthesis in detail better involves functional knockouts of domains, or domain depletion and duplications, which has been performed for such systems before (see for example Hayashi et al. 2016 Sci Rep, for analyzing the impact of ACP domains). I recommend refraining from extracting any deeper information from this dataset, but rather stressing the technological relevance of shifting product spectra by chimeric BGCs.

We agree with reviewer #2: Our experimental set-up is too complex to allow specific statements regarding the molecular basis of PUFA synthesis. We therefore relativized the statements regarding the impact of our approach for the understanding of the mechanistic insights into PUFA synthases and our conclusions drawn on the function of the DH domains. Additionally, we indeed stressed the

biotechnological relevance of shifting product spectra by hybrid PUFA synthases for producing specific PUFAs.

Minor points:

(1) Although the approach is clear and experiments thoroughly described, the improved figures and a different arrangement of figure panels would make the manuscript better readable. For example, Figure 2a shows the BGC of *A. fasciculatus*. PUFA data to this BGC in *Y. lipolytica* is shown in Figure 2c together with data from the second BGC (*Minicystis rosea*). The *Minicystis rosea* BGC is, however, introduced much later. It is very difficult to read data in Figure 2c when the information to the second BGC is withheld several pages. Please rearrange this chapter or the figure panels. In addition, the squares for showing the color code are too small and colors are too similar. Figure legend also misses label "c". I also recommend showing structures of the main polyunsaturated fatty acids produced by myxobacteria, and recommend improving Figure 1 by including AGPAT in the reaction scheme. Any information that is available on the interplay of pfa1, 2 and 3 should be included as well. According to the reviewer's suggestions, the text was rearranged, the squares for showing the color code in Fig. 2c were magnified and figures 2 and 3 were transformed from grayscale figures into color figures. The colors are now easy to distinguish. Moreover, the structures of the main polyunsaturated fatty acids produced by the myxobacteria *Aetherobacter fasciculatus* and *Minicystis rosea* as well as the reaction catalyzed by AGPAT are now included in Fig. 1. The legend of Fig. 1 was adapted and the catalytic domains are now correlated with the respective Pfa protein.

(2) The nomenclature of the strains/plasmid is very difficult. It needed the printed figures next to the screen to be able to decode the results section and to connect data to constructs. Please think of a simpler nomenclature? There are 11 different clusters shown. It may be possible to assign a specific color to each of this cluster, and than color code data presentation. All this would make the paper better readable.

The long names of plasmids and strains were shortened according to the reviewer's suggestion. All the gene clusters shown in Figures 2 and 3 were transformed from grayscale figures into color figures. A specific color code is now assigned to the 11 different clusters and to the corresponding production profile shown in Fig. 3.

(3) For comparison, include data of wildtype-like BGCs (shown in Figure 2) in Figure 3.

The PUFA production profile of strains *Po1h::SynPfaPptAf4* and *Po1h::SynPfaPptMr2*, containing wildtype-like BGCs, are now also included in Fig. 3.

(4) Can parts be omitted from the main text? All data to codon harmonization are presented in the SI. Also the chapter to codon harmonization could be moved from the main text to SI. Details to construct Ppt7 could be moved to SI, too. A sentence in the main text saying that ER exchanges do not influence the product spectrum should be enough.

The chapter “Design of a new version of an artificial DPA/DHA-type pfa gene cluster and heterologous LC-PUFA production” and the corresponding chapters in the “Methods” section were moved from the main text to the SI accordingly. This led to a substantial shortening of the main text.

The work of Gemperlein et al. surely warrants publication. Although the approach is not highly novel (see Gemperlein 2014 Chem Sci, and Gemperlein et al. 2016 Metabolic Engineering), the study demonstrates the successful production of PUFAs at yet unmatched titers and variability. The suitability of *Y. lipolytica* for production of PUFAs, a thorough experimental set-up as well as the huge experimental effort makes this study a success, which is worth sharing with the readership of Nature Communications.

REVIEWERS' COMMENTS:

Reviewer #1 (Remarks to the Author):

The manuscript is much improved. By taming their claims, it is more acceptable to not provide mechanistic explanations. Therefore the manuscript is acceptable for publication.

Reviewer #2 (Remarks to the Author):

The data presented by Gemperlein et al. are highly interesting. Unfortunately, the original manuscript suffered from overloading with data, a lengthy nomenclature of strains and a partly poor explanation of data in both text and figure. Both reviewers have essentially addressed these points. The authors have carefully revised the manuscript. It is now slimmer, allowing to present the results much clearer and to highlight the key aspects of their work. This manuscript is of similar impact than many of recent manuscripts published in Nature Communications on the microbial production of technologically relevant platform chemicals. I am therefore convinced that the manuscript of Gemperlein et al. deserves publication in Nature Communications.

We thank the reviewers for their remarks. There are no further issues raised by the reviewers.

Reviewer #1 (Remarks to the Author):

The manuscript is much improved. By taming their claims, it is more acceptable to not provide mechanistic explanations. Therefore the manuscript is acceptable for publication.

Reviewer #2 (Remarks to the Author):

The data presented by Gemperlein et al. are highly interesting. Unfortunately, the original manuscript suffered from overloading with data, a lengthy nomenclature of strains and a partly poor explanation of data in both text and figure. Both reviewers have essentially addressed these points. The authors have carefully revised the manuscript. It is now slimmer, allowing to present the results much clearer and to highlight the key aspects of their work. This manuscript is of similar impact than many of recent manuscripts published in Nature Communications on the microbial production of technologically relevant platform chemicals. I am therefore convinced that the manuscript of Gemperlein et al. deserves publication in Nature Communications.